# Simulating out-of-sample atmospheric transport to enable flux inversions

Nikhil Dadheech[1] and Alexander J. Turner[1]

[1]Department of Atmospheric and Climate Science, University of Washington, Seattle, WA, USA

**Correspondence:** Alexander J. Turner (turneraj@uw.edu)

**Abstract.** Accurately estimating greenhouse gas (GHG) emissions from atmospheric observations requires resolving the up-wind influence of measurements via atmospheric transport and dispersion models. However, the computational demands of physics-based models limit the scalability of flux inversions, particularly for dense in situ and satellite-based observations. Here, we present FootNet v3, a deep-learning emulator of atmospheric transport based on a U-Net++ architecture, which improves generalization and inversion fidelity over prior U-Net-based models. FootNet v3 is trained on 500,000 footprint examples across the contiguous United States. It predicts surface and column-averaged source-receptor relationships at kilometer-scale resolution and operates $650\times$ faster than traditional Lagrangian models. Critically, FootNet learns the underlying physical relationship between meteorology and atmospheric transport. We show that it produces consistent source-receptor relationships when driven by GFS meteorology, even though it was trained with HRRR inputs. FootNet generalizes to unseen regions and meteorological regimes, enabling accurate flux inversions in domains withheld during training. Case studies using GHG measurements in the San Francisco Bay Area and Barnett Shale show that FootNet matches or exceeds the performance of physics-based models when used in a flux inversion and evaluated against independent GHG observations. This is achieved despite FootNet having never seen meteorological inputs from Northern California or North Texas. Feature importance testing identifies physically meaningful drivers that are consistent across both surface and column models. These findings show that machine learning models can learn the underlying physical relationships governing atmospheric transport, allowing them to extrapolate to out-of-sample scenarios and support real-time, high-resolution GHG flux estimation in novel domains without the need for retraining or precomputed footprint libraries.

## 1 Introduction

Carbon dioxide ($CO_2$) and methane ($CH_4$) are the two most important greenhouse gases (GHGs). Together, they account for more than 85% of the cumulative radiative forcing since the pre-industrial era (IPCC, 2023). Accurate characterization of their sources and sinks across spatial scales is essential for constraining future climate trajectories. However, the computational burden and data storage demands of physics-based atmospheric transport models limit the scalability of current inversion frameworks, especially when leveraging dense GHG observations (Roten et al., 2021; Cartwright et al., 2023; Fillola et al., 2023, 2025). To overcome these limitations, He et al. (2025) and Dadheech et al. (2025) introduced FootNet, a machine learn-ing emulator of atmospheric transport tailored for surface observations. Here, we extend FootNet into a generalized framework

(FootNet v3) trained over the Contiguous United States (CONUS), enabling the emulation of both surface and column-averaged source-receptor relationships ("footprints") at kilometer-scale resolution. Crucially, we demonstrate that FootNet v3 generalizes to previously unseen regions and meteorological conditions, enabling accurate out-of-sample simulation of atmospheric transport. This capability represents a significant advance, as it addresses a long-standing limitation in inversion systems: the reliance on site-specific, computationally intensive modeling.

Recent work has underscored the value of dense, continuous GHG observations for quantifying and attributing emissions (e.g., Turner et al., 2020; Varon et al., 2023; Hamilton et al., 2024; Asimow et al., 2024, 2025). Observation networks such as INFLUX (Davis et al., 2017) and BEACO$_2$N (Shusterman et al., 2016) have expanded the spatial and temporal resolution of in situ $CO_2$ and methane measurements, particularly in urban environments. Concurrently, satellite-based retrievals have advanced dramatically: OCO-2 and OCO-3 provide column-averaged $CO_2$ at 2.25 km $\times$ 1.29 km resolution on 16-day cycles (O'Dell et al., 2012; Eldering et al., 2019), TROPOMI offers daily methane retrievals at 5.5 km $\times$ 7 km (Veefkind et al., 2012), and MethaneSAT will resolve emissions at 130 m $\times$ 400 m (Rohrschneider et al., 2021).

GHG inversion frameworks relate atmospheric observations to surface fluxes through a linear transport operator:

$$\boldsymbol{y} = \mathbf{H}\boldsymbol{x} + \boldsymbol{b}, \tag{1}$$

where $\boldsymbol{y}$ is the vector of $n$ observations, $\boldsymbol{x}$ the vector of $m$ fluxes, $\boldsymbol{b}$ the background, and $\mathbf{H}$ the Jacobian matrix describing transport. Each row of $\mathbf{H}$ encodes the influence of surface fluxes on a specific observation, and each column reflects the impact of a surface pixel on all observations. As the resolution of $\boldsymbol{x}$ or density of $\boldsymbol{y}$ increases, the cost of computing and storing $\mathbf{H}$ becomes prohibitive.

Physics-based models such as Eulerian transport solvers or Lagrangian Particle Dispersion Models (LPDMs) are commonly used to construct $\mathbf{H}$, which is then used to infer GHG fluxes within an inversion framework. For example, the Integrated Methane Inversion (IMI) is an Eulerian-based inversion framework focusing on regional scale at 25 km resolution (Varon et al., 2022; Estrada et al., 2024). Eulerian-based frameworks struggle with high spatiotemporal resolutions as the number of model simulations is proportional to the dimensions of $\boldsymbol{x}$. Variational methods such as 4D-var are also popular in flux inversions, which require an adjoint of the Eulerian model to compute the atmospheric transport (Henze et al., 2007). LPDMs like STILT and X-STILT are often preferred at high resolution due to their flexibility and ability to resolve localized sensitivity patterns (Lin et al., 2003; Fasoli et al., 2018; Wu et al., 2018). However, these models scale linearly with the number of observations, creating a computational bottleneck for dense observational datasets.

Several recent studies have explored the use of machine learning to emulate LPDM outputs and reduce computational costs (Roten et al., 2021; Cartwright et al., 2023; Fillola et al., 2023). These approaches typically interpolate or approximate LPDM simulations using learned surrogate models, but they often remain dependent on precomputed LPDM libraries or perform poorly outside of the training domain. FootNet eliminates this dependency by enabling direct inference of atmospheric transport sensitivities without additional LPDM simulations. He et al. (2025) first demonstrated this approach for surface footprints in two domains; Dadheech et al. (2025) improved near/far-field balance and showed that FootNet could support high-resolution urban flux inversions. Here, we substantially broaden the scope of the model. FootNet v3 is trained on 500,000 footprint

examples spanning a wide range of locations, seasons, and meteorological conditions across CONUS. It comprises separate models for surface and column-averaged footprints and is trained using STILT and X-STILT outputs, respectively.

We emphasize a key result: FootNet v3 enables out-of-sample simulation of atmospheric transport. We evaluate its generalizability using flux inversion case studies in domains withheld from training, including $CO_2$ inversions in the San Francisco Bay Area and methane inversions over the Barnett Shale. In both cases, FootNet v3 matches or outperforms physics-based LPDMs. This result demonstrates, for the first time, that machine learning can replicate transport model performance in novel regions without re-running expensive simulations. The ability to emulate atmospheric transport out-of-sample is a foundational step toward operational, real-time flux inversions at continental scale.

## 2 Development of a generalizable machine learning emulator for atmospheric transport

FootNet v3 adopts a U-Net++ architecture (Zhou et al., 2018) with 37M parameters, replacing the U-Net architecture used in earlier versions (He et al., 2025; Dadheech et al., 2025). Figure 1 shows a schematic of the architecture. The U-Net++ is a deep encoder-decoder model designed to preserve multiscale spatial features while improving feature fusion between encoder and decoder layers. This nested, dense skip-connection design reduces the imbalance between near and far-field footprint sensitivity observed in prior versions. We trained two separate models within this framework: one for surface footprints and one for column-averaged footprints. Both models share the same general architecture but differ in their input feature sets to reflect the physical drivers of transport relevant to surface and column observations.

To train FootNet v3, we constructed a training dataset of 500,000 footprints across the CONUS domain to support generalizable inference of transport sensitivity. Figure 2 illustrates the spatial receptors sampling strategy, which consists of two components: (1) uniformly distributed individual receptors (gray dots) and (2) 400 km $\times$ 400 km subdomains with enhanced sampling (red stars). A receptor refers to a specified location, time, and measurement type (surface or column-averaged) for which a corresponding footprint is simulated. Each receptor is randomly sampled across months, days, and hours in 2020 and 2021 to expose the model to diverse meteorological states. In the enhanced sampling regions, we generated approximately 2,500 footprints per domain, capturing local variation in winds, boundary layer dynamics, and terrain.

These 500,000 footprints were split into two training configurations. The first, which we refer to as the *out-of-sample FootNet*, excludes all data from 2020 and two regions: the bulk of California and much of Texas. The out-of-sample FootNet uses footprints from 2021 for training. Within these two regions are two case studies reserved for later out-of-sample evaluation: the San Francisco Bay Area (Domain A) and the Barnett Shale region (Domain B). These out-of-sample evaluations are done using the observations from 2020. The second configuration, the *in-sample FootNet*, includes all 500,000 observations from both 2020 & 2021, and provides a high-capacity emulator suitable for deployment. For validation and testing, we randomly picked 400$\times$400 km$^2$ subdomains (similar to Figure 2) with enhanced temporal sampling, as well as footprints computed for randomly sampled receptors across CONUS that do not correspond to the training receptor locations. We used 50,000 footprints randomly sampled in 2020 and 2021 for testing and validation.

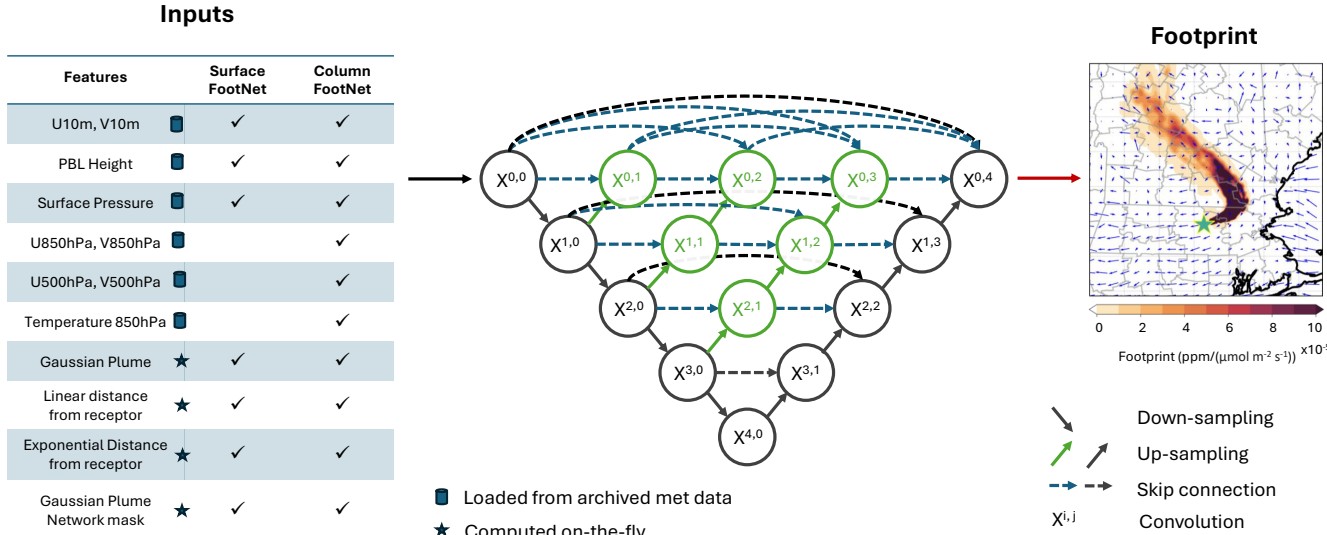

**Figure 1.** Schematic of the UNet++ architecture used in FootNet v3. The green circles and green & blue arrows represent new layers and their respective connections compared to the earlier U-Net architecture. Input features include meteorological fields, Gaussian plumes, and spatial context features. The model outputs the predicted footprint sensitivity for a given observation. All meteorological inputs include fields from 0hr, 6hr, 12hr, 18hr, and 24hr before the receptor time. We applied a $3\times3$ convolution kernel on the convolution layers. The U-Net++ architecture is adapted from Zhou et al. (2018).

Each receptor was simulated using STILT (Lin et al., 2003; Fasoli et al., 2018) for surface footprints and X-STILT (Wu et al., 2018) for column-averaged footprints, using NOAA High Resolution Rapid Refresh (HRRR) meteorology at 3 km resolution regridded to 1 km. The trajectories were simulated backward in time for 72 hours, or until the particles exited the domain, whichever occurred first. Surface and column simulations were co-located in space and time, providing a matched training set across footprint types. The result is a comprehensive dataset enabling FootNet to learn robust mappings from meteorological inputs to transport sensitivities across CONUS.

As mentioned above, we trained two models using the same general architecture: one for surface footprints and one for column-averaged footprints. Inputs to each model include meteorological fields interpolated from HRRR to 1 km resolution. This is done to ensure that all the input fields (e.g., meteorology, Gaussian plume, etc.) are at the same spatial resolution. These meteorology fields include zonal and meridional wind components, temperature, surface pressure, and boundary layer height sampled at 0hr, 6hr, 12hr, 18hr, and 24hr before the receptor time (i.e., backward in time). We found minimal improvement in performance when including meteorological inputs more than 24 hours backward in time. To guide spatial structure, we include the linear distance to the receptor and a simplified Gaussian plume estimate derived from surface winds. Importantly, many of these fields do not need to be stored on disk and can be computed through a few floating point operations. The "Inputs" table in Figure 1 indicates the fields that must be loaded from disk and which are computed on-the-fly. Column FootNet

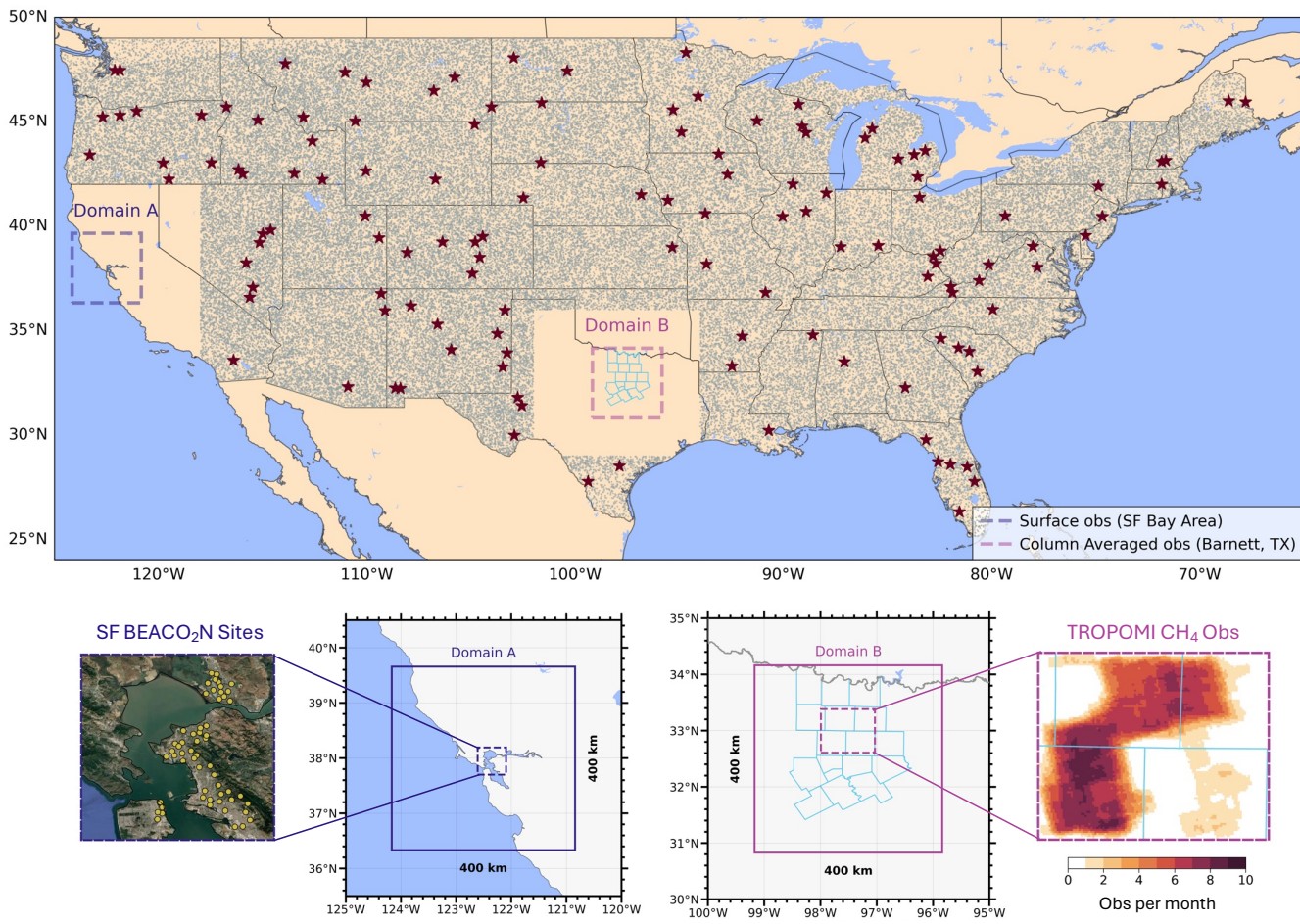

**Figure 2.** (Top panel) Receptors for training out-of-sample FootNet. Gray dots indicate individual receptors sampled uniformly across CONUS. Red stars show centers of $400 \times 400$ km$^2$ subdomains with enhanced temporal sampling. Domains A and B are withheld in the out-of-sample training configuration. Supplemental Figure S1 shows the full training data used for the in-sample FootNet model. Bottom row shows the observational networks used in both case studies. The blue boundaries in Domain B show counties in the Barnett Shale basin. Background satellite imagery is taken from Google Maps.

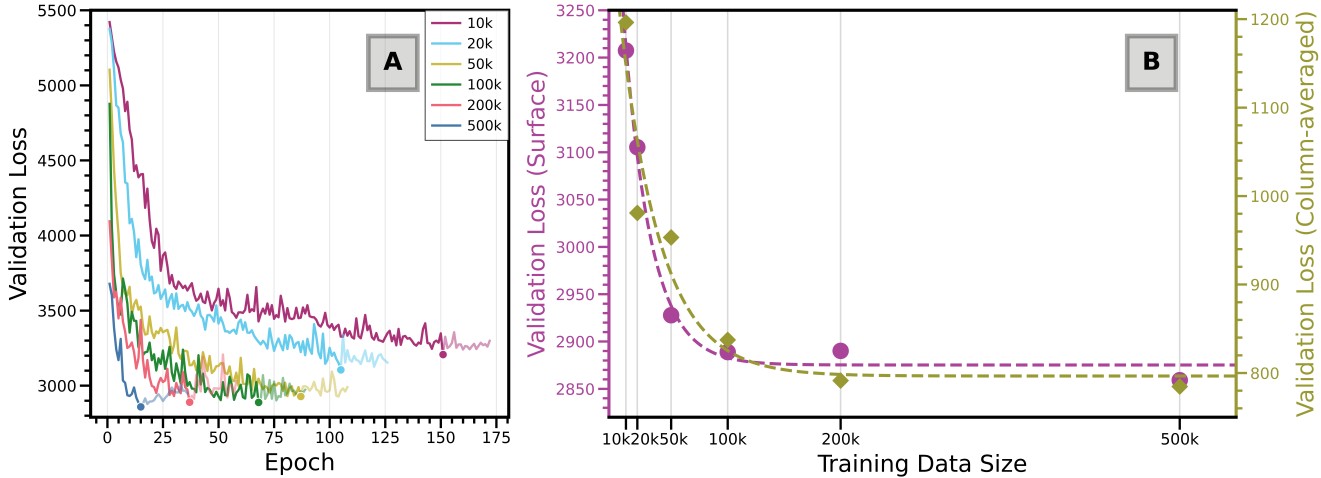

**Figure 3.** Training set size versus model performance. (A) Evolution of validation loss for surface footprint models trained with different dataset sizes. (B) Minimum validation loss for surface and column models as a function of training set size. Results shown are for a single random seed.

models additionally incorporate winds at 850 and 500hPa, and temperature at 850hPa. These features enable the model to learn sensitivity to both near-surface and free-tropospheric transport. The model outputs a footprint field matching the domain of the meteorological input grid. We use mean squared error as the primary loss function, with an optional penalty on total footprint mass to encourage mass conservation (mass conservation is described later).

The size of the full training data set is 30 TB. Given the scale of the training set, we distributed training across 18 NVIDIA A2 GPUs (16GB memory) using PyTorch's Distributed Data Parallel (DDP) framework. Each GPU processes a distinct shard of the data, with gradients synchronized through NVIDIA's NCCL backend. This reduced training time from several months on a single GPU to under 10 days. Once trained, FootNet can compute a footprint in under a second on a single GPU. This represents a $650\times$ speedup relative to the physics-based model and enables near-real-time inference in flux inversion workflows.

    An important question we encountered while generalizing FootNet over the whole CONUS was *"how many training samples do we need to ensure generalizability of the FootNet model?"* To quantify how training set size impacts model skill, we trained multiple versions of FootNet using subsets of the full 500,000-sample dataset. Each version was trained from scratch with the same architecture, loss function, and training protocol. Figure 3 summarizes the evolution of validation loss over training

epochs and the minimum loss achieved for each subset size. Figure 3a shows that models trained on smaller datasets converge more slowly and to higher final loss values. Increasing the number of training samples yields consistent gains in predictive skill up to roughly 100,000 examples. Beyond this point, improvements begin to saturate, suggesting diminishing returns on validation loss for additional data. This asymptotic behavior is consistent for both surface and column-averaged footprint models (Figure 3b). For generalizable inference with uniform skill across the diverse meteorological and geographic conditions

represented within CONUS, the full training set remains essential.

Mass conservation is a defining feature of physical transport models, and its absence can lead to artifacts in source-receptor relationships. This may be particularly problematic when used in a flux inversion as one may infer erroneous fluxes that do not conserve mass. While traditional LPDMs enforce conservation through explicit particle tracking and boundary layer diagnostics, machine learning models such as FootNet lack built-in physical constraints. Here, we explore whether a simple regularization strategy can help enforce mass conservation without degrading predictive skill.

The footprint value for a pixel at a given timestep is directly proportional to the mass of the air parcel (Lin et al., 2003). As such, we modified the loss function used during training to include a penalty on the difference in total mass between the $i^{\text{th}}$ predicted footprint $\mathbf{h}_i$ and the reference footprint $\hat{\mathbf{h}}_i$ from STILT or X-STILT:

$$\mathcal{L}(\mathbf{h}_i, \hat{\mathbf{h_i}}) = \text{MSE}(\mathbf{h}_i, \hat{\mathbf{h_i}}) + \alpha \frac{\left| \sum \mathbf{h}_i - \sum \hat{\mathbf{h}}_i \right|}{\sum \hat{\mathbf{h}}_i}, \tag{2}$$

where $\alpha$ is a tunable penalty weight and $\text{MSE}$ is the mean squared error. The second term acts as a soft constraint, discouraging total footprint biases without imposing strict conservation.

We conducted a sensitivity analysis to assess how this constraint affects performance. Figure 4 shows the trade-off between validation loss (MSE) and percentage footprint sum difference for different values of $\alpha$. For surface footprints, modest penalty values (i.e., $\alpha = 1000$) reduce mass conservation errors without significantly increasing validation loss. Larger values of $\alpha$ overly constrain the network and degrade performance. For column footprints, we found no clear benefit to including a mass penalty, and the final column model was trained with $\alpha = 0$.

Although this penalty is not a substitute for explicit mass-tracking, it offers a lightweight and computationally efficient way to discourage unphysical footprint predictions. Other approaches for mass conservation could be explored in future work (e.g., Sturm and Wexler, 2020; Wang and Gupta, 2024; Meng et al., 2025). This approach may be especially useful in applications where integrated footprint magnitude directly affects inversion results.

To evaluate how well FootNet generalizes across space and observation type, we compare predicted footprints against STILT and X-STILT outputs in regions and conditions not used during training. Figure 5 shows representative examples drawn from the independent test set, including surface and column-averaged footprints across a range of meteorological states. In each case, FootNet captures the dominant spatial structure of the reference footprint, including the directionality imposed by wind fields and the localization associated with boundary layer mixing. For example, in Massachusetts, the model reproduces a classic Gaussian plume aligned with surface winds for a hypothetical surface observation. In Michigan, we can see that the X-STILT column footprint hugs the shoreline of Lake Michigan and FootNet reproduces this complex spatial pattern with high fidelity. Similarly, a column example from Utah shows a complex footprint shaped by topography and mesoscale flows, which is well-approximated by FootNet. In Texas, the large-scale flow is consistent between FootNet and STILT, but STILT exhibits sharper, more localized structures, whereas FootNet yields a smoother footprint. While STILT's spatial structure is more physically *realistic* (as it directly solves the governing equations), it is not necessarily more *accurate* due to potential biases in the driving meteorological fields. Highly localized but biased footprints could introduce artifacts in GHG flux inversions. In this context, the smoother prediction from FootNet may actually be preferable. However, it is important to note that FootNet is trained on STILT and X-STILT. As such, the improved performance against independent atmospheric observations implies that the

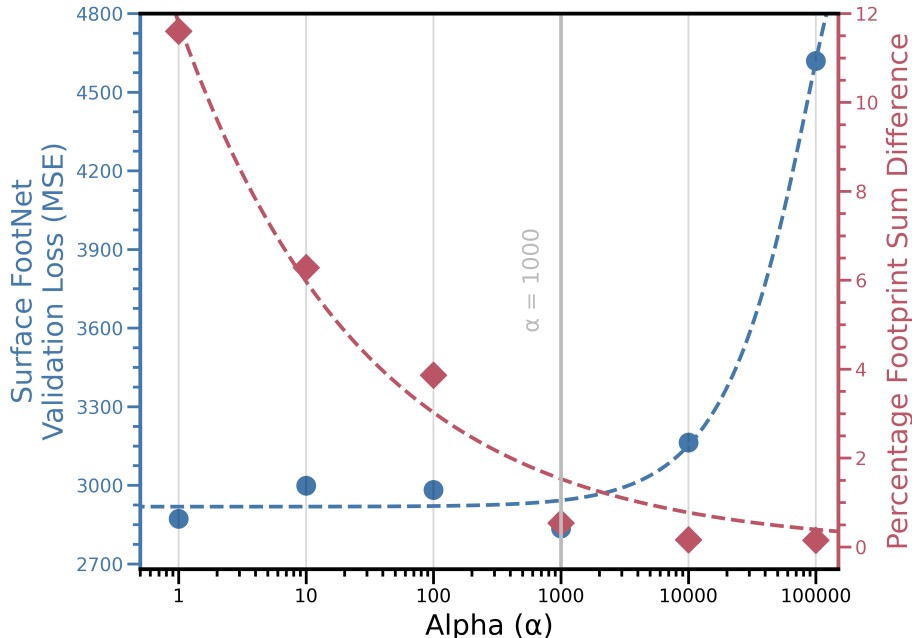

**Figure 4.** Trade-off between validation loss (mean squared error; MSE) and percentage footprint sum difference for different values of $\alpha$ in the surface FootNet model.

smoothness is mitigating underlying model errors. We also note that other approaches have been developed to mitigate transport errors within physics-based models and inversion frameworks. For example, incorporating stochastic wind uncertainties to enhance dispersion (Lin and Gerbig, 2005), and explicitly accounting for correlated transport errors over the footprint duration (Jones et al., 2021). Overall, Figure 5 shows that FootNet reproduces footprints with high fidelity across a wide range of conditions.

As noted in Dadheech et al. (2025) and can be seen in Figure 5, FootNet outputs are often smoother than those from LPDMs, which can exhibit sharp boundaries due to discrete particle trajectories. Machine learning models generalize by finding an underlying trend instead of fitting every noisy data point (Shukla et al., 2021). FootNet model architecture consists of Convolutional Neural Network (CNN) layers that perform convolution operations on local neighborhoods instead of processing individual pixels (Bishop, 2006). These are the two primary reasons why the FootNet model produces smoother outputs. While

this smoothing can obscure fine-scale structure, it reduces sensitivity to meteorological noise and can improve stability in inversion settings. These results demonstrate that FootNet reliably reproduces transport patterns across diverse regions and observation types. This generalization is essential for scaling inversion systems to new domains without re-running expensive transport simulations.

   To assess FootNet's sensitivity to the meteorological inputs used in training and evaluation, we tested its ability to gener-

ate footprints using meteorology from the Global Forecast System (GFS), which differs substantially from the HRRR fields

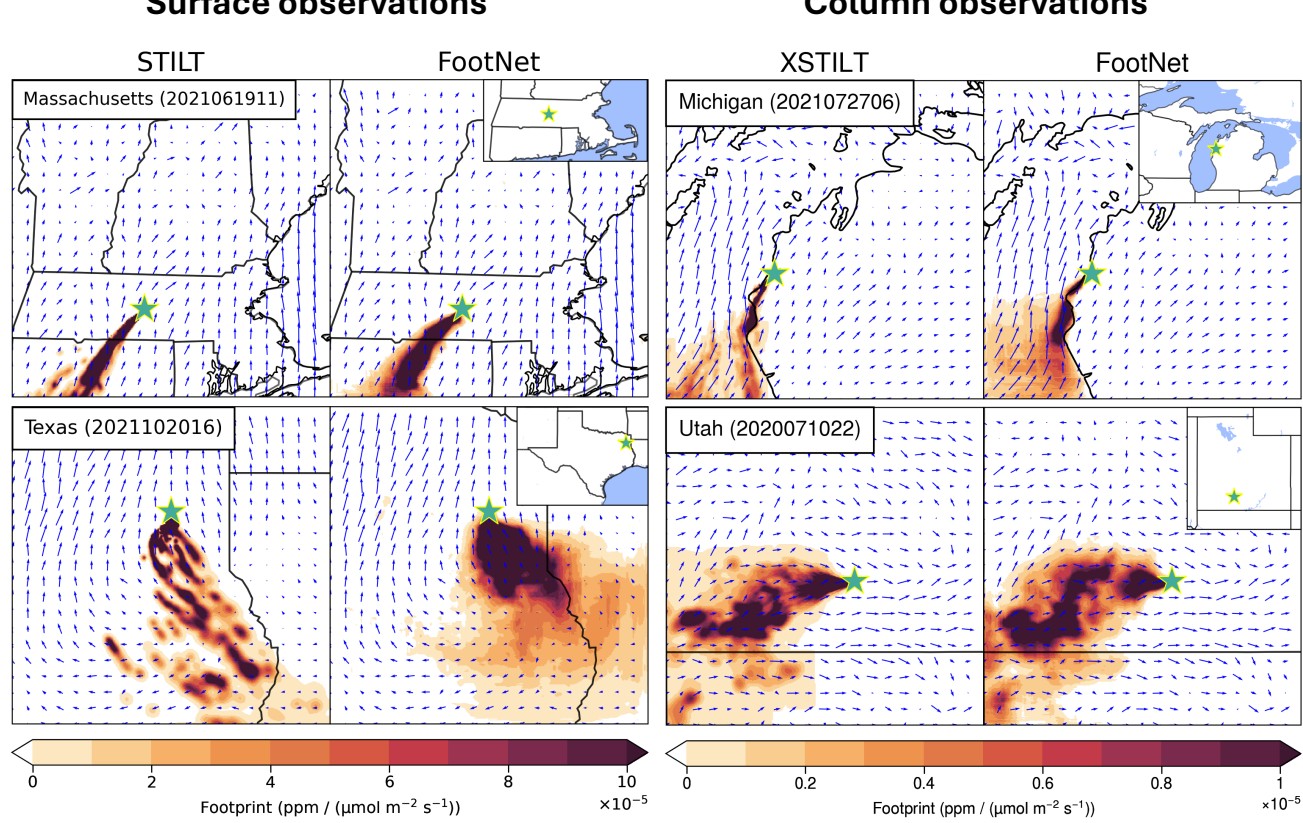

**Figure 5.** Comparison of in-sample FootNet-predicted footprints (right column) with STILT or X-STILT footprints (left column) for randomly selected test examples. Green stars mark receptor locations; blue arrows show instantaneous wind direction. Examples span both surface and column footprints across multiple CONUS regions. Figure S7 shows a similar comparison for out-of-sample FootNet-predicted footprints for the year 2020.

on which FootNet was trained. Specifically, we compared footprints generated by FootNet using GFS meteorology to STILT footprints computed with GFS meteorology. Figure 6 shows footprint comparisons for the same receptors in the San Francisco Bay Area. We note that these footprints were withheld from the FootNet training process. FootNet predictions using HRRR meteorology closely match STILT-HRRR footprints, as expected. Notably, higher bias occurs for small footprint values, particularly in the far-field, where the smoothness of FootNet footprints results in deviations from the ground truth STILT footprints. When driven with GFS meteorology, FootNet footprints still capture the overall structure and magnitude of the STILT-GFS footprints, despite being trained using meteorology from HRRR. These results suggest that FootNet is learning the underlying physical relationship between meteorology and the source-receptor relationship.

This further suggests that while FootNet is optimized for the meteorology on which it is trained, it maintains skill when driven by alternate meteorological products. This is important because HRRR is only available over CONUS, whereas GFS is a global product. This means that FootNet can be used in domains outside of CONUS with GFS or any other global meteorology products. Given the known discrepancies between meteorological models and among LPDMs themselves, these differences seen in Figure 6 fall within expected tolerances. For instance, STILT and FLEXPART can produce similar or larger disagreements than observed here when run under similar conditions (Karion et al., 2019; Munassar et al., 2023). This flexibility allows FootNet to support flux inversions using multiple sources of meteorology, including global reanalysis products, and implies that FootNet may perform well in simulating atmospheric transport in out-of-sample domains.

## 3 Evaluating in-sample and out-of-sample flux inversion performance

A central aim of this study is to assess whether FootNet can enable high-resolution flux inversions in regions excluded from training. To test this, we conducted flux inversion experiments using both surface and column observations in two held-out domains: the San Francisco Bay Area (Domain A) and the Barnett Shale region in Texas (Domain B). For each domain, we performed three inversions using: (1) a physics-based LPDM (baseline), (2) an in-sample FootNet model trained on all of CONUS, and (3) an out-of-sample FootNet model trained with data from 2021 and the target domain withheld (see Fig. 3). These experiments evaluate the generalizability of FootNet in practical inversion settings. The in-sample model was trained on 500,000 samples, including data from Domains A and B, while the out-of-sample models excluded data from both of the evaluation domains. Demonstrating accurate inversions under this setup is critical for scaling flux estimation frameworks to other observation networks.

For Domain A (San Francisco Bay Area), we used hourly $CO_2$ observations from the BEACO$_2$N network between February and May 2020. Inversions used three footprint configurations: STILT (baseline), in-sample FootNet, and out-of-sample FootNet (with no Bay Area training data). All inversions used the same Bayesian framework and prior fluxes, based on previous work (McDonald et al., 2014; Turner et al., 2016, 2020; Dadheech et al., 2025). For Domain B (Barnett Shale), we performed daily methane inversions using TROPOMI column data between February and April 2020. The same three configurations were applied: X-STILT (baseline), in-sample FootNet, and out-of-sample FootNet (excluding Barnett Shale data). All inversions used the same Bayesian framework and prior fluxes, with the Environmental Protection Agency (EPA) anthropogenic methane

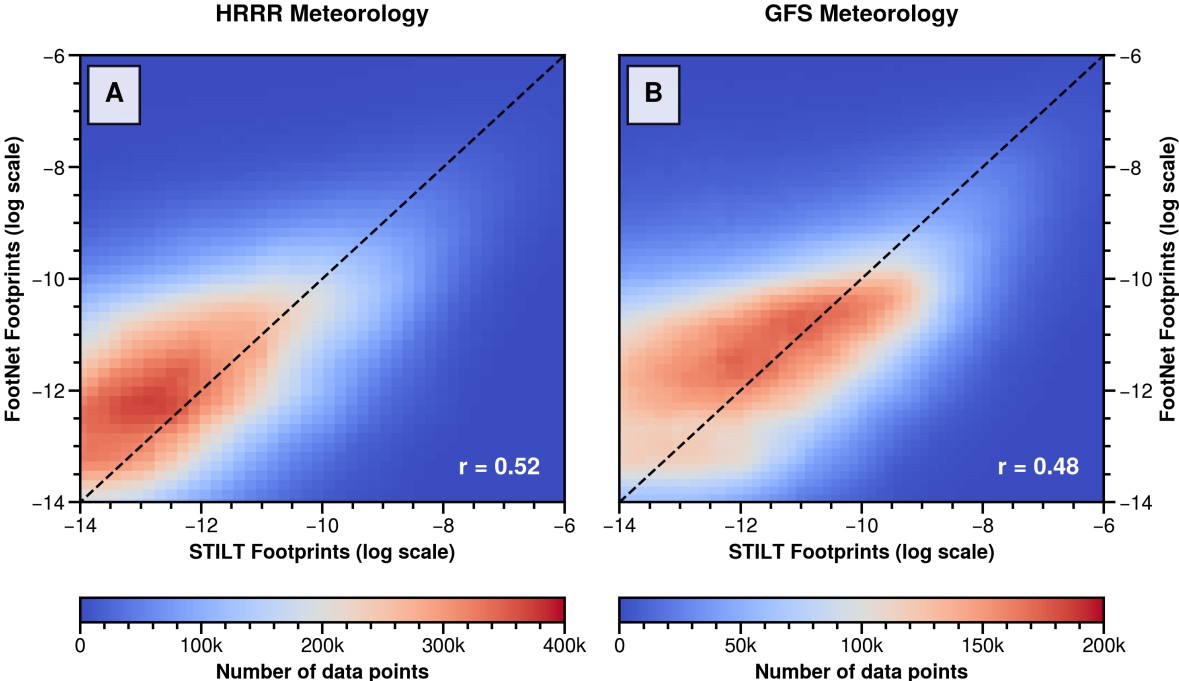

**Figure 6.** Comparison between FootNet and STILT footprints using different meteorological inputs. The x-axes show STILT-simulated footprint values using HRRR (left) and GFS (right) meteorology. The y-axes show corresponding FootNet predictions using the same meteorological inputs. FootNet was trained exclusively on HRRR data and has never seen GFS meteorology. All predictions are made at the same receptor locations and compared against independent test footprints withheld from training. Each data point in the plot represents a non-zero influence pixel from one of the 5000 randomly sampled footprints in the test set. The unit of the footprints is ppm/($\mu$mol m$^{-2}$ s$^{-1}$).

emission inventory as the prior (Maasakkers et al., 2023). A detailed description of the inversion setup for Domains A & B is provided in Appendix A.

Figures 7 and 8 show inversion performance against $CO_2$ and methane observations withheld from the inversions. All statistics are computed using these independent observations. The out-of-sample FootNet setup (bottom rows in Figs. 7 and 8) evaluates model performance in regions entirely excluded from training, constituting a rigorous test of spatial generalization. Unlike previous work that included training data from the San Francisco Bay Area, albeit for different time periods (Dadheech et al., 2025). This configuration directly probes FootNet's extrapolation capability.

Figure 7 summarizes the Bay Area inversion results. Both in-sample and out-of-sample FootNet models produce smoother footprints than STILT, leading to broader cumulative influence. Consistent with the findings of Dadheech et al. (2025), FootNet outperforms STILT when compared against independent $CO_2$ observations. The key distinction here is that FootNet achieves this performance even when trained exclusively on data outside of the region. All three configurations identify high emissions along freeways and around the Bay. Overall, the FootNet models exhibit better correlation and lower mean squared error

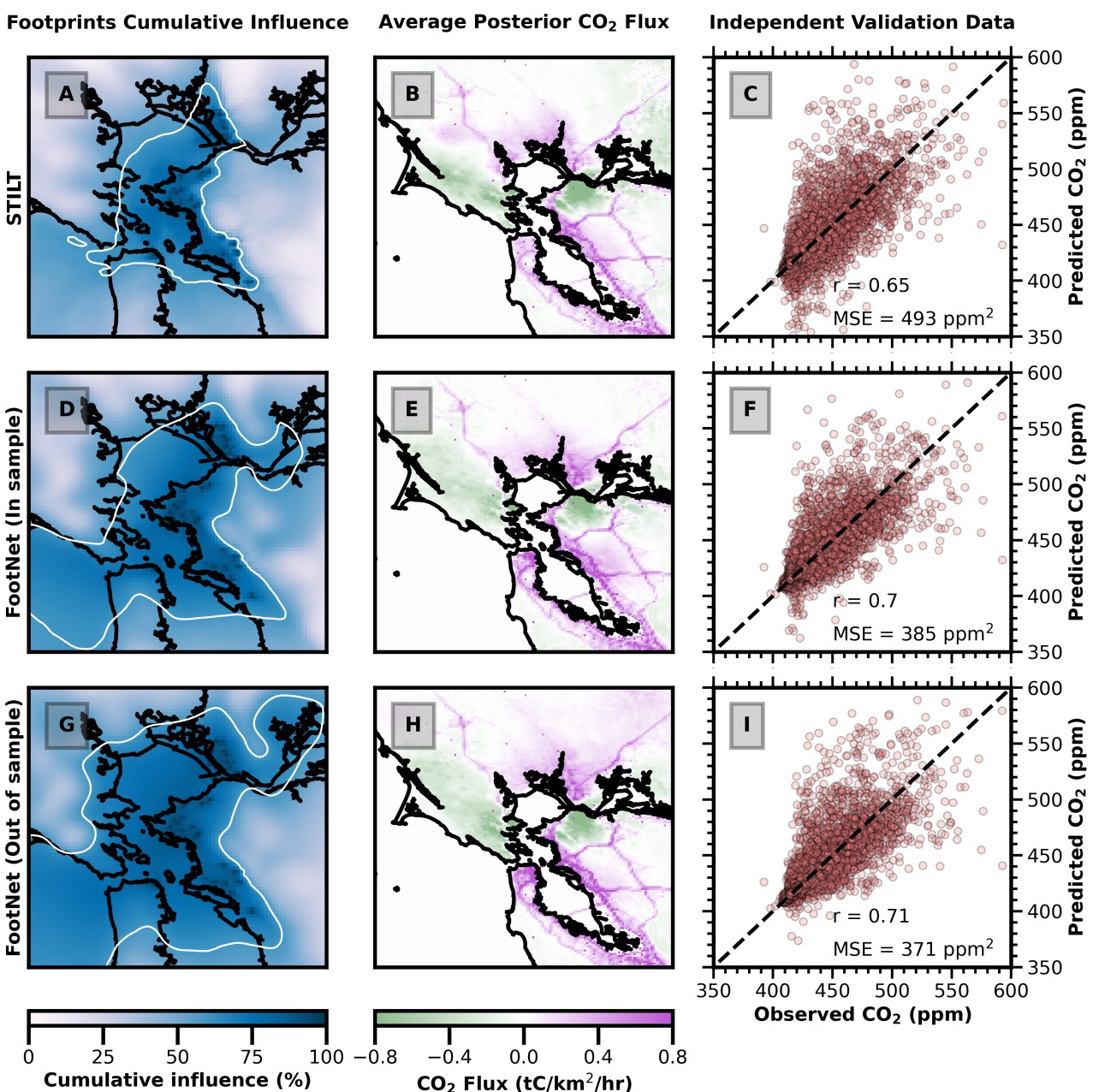

**Figure 7.** $CO_2$ inversion results in the San Francisco Bay Area using STILT, in-sample FootNet, and out-of-sample FootNet footprints. (Left column) Cumulative footprint influence. (Middle column) Posterior $CO_2$ flux inferred from the BEACO$_2$ network. (Right column) Comparison of observed and simulated $CO_2$ concentrations from an independent test set using the corresponding posterior fluxes. Observations were drawn using a consistent seed across experiments. White contours in the left column represent the $60^{th}$ percentile of the cumulative footprint influence. The percentage cumulative influence is relative to the cumulative footprint sum for each grid point.

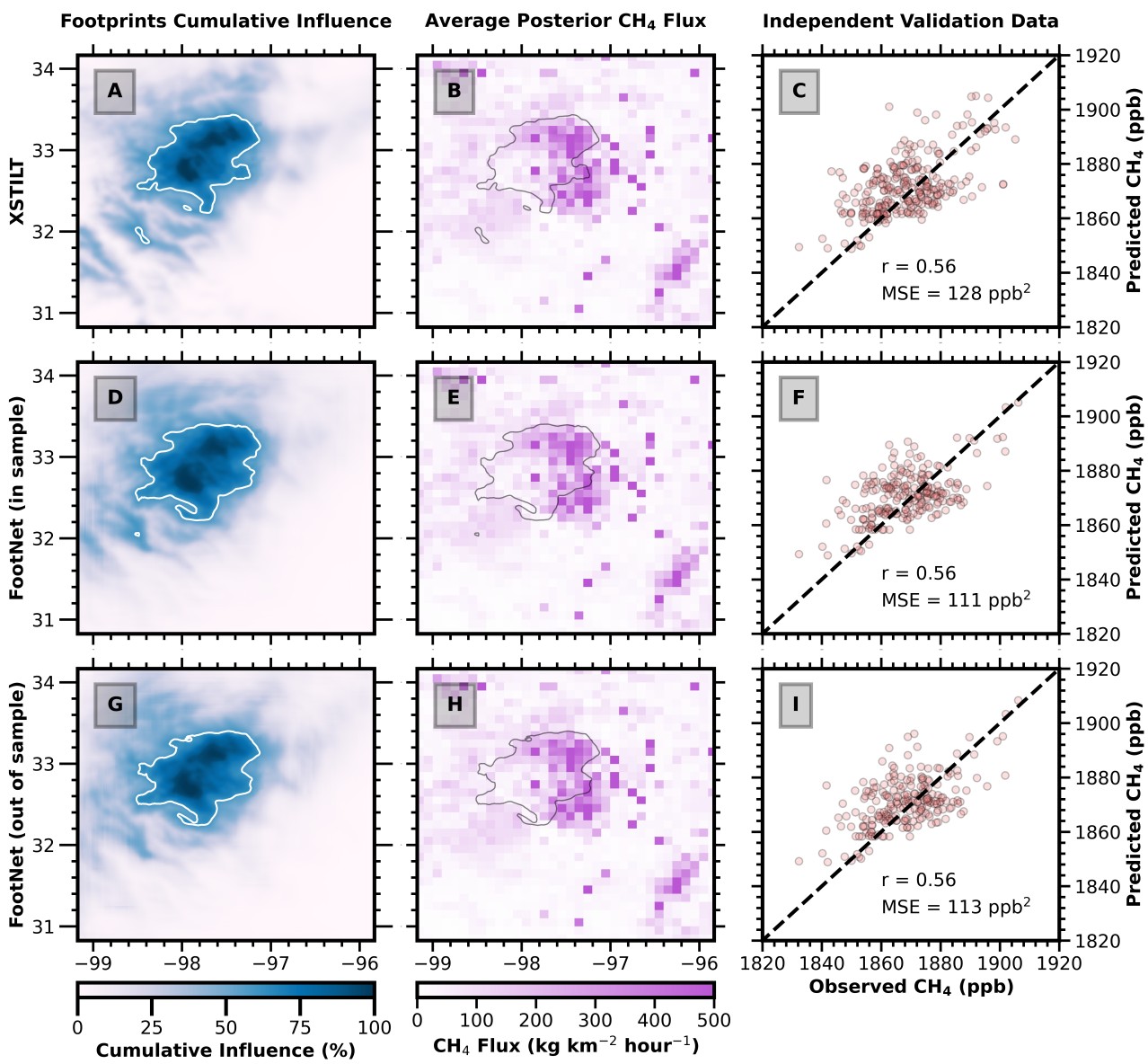

**Figure 8.** Methane inversion results in the Barnett Shale, TX using X-STILT, in-sample FootNet, and out-of-sample FootNet footprints. (Left column) Cumulative footprint influence. (Middle column) Posterior methane flux inferred from the TROPOMI observations. (Right column) Comparison of observed and simulated methane concentrations from an independent test set using the corresponding posterior fluxes. Observations were drawn using a consistent seed across experiments. White contours in the left column represent the $60^{th}$ percentile of the cumulative footprint influence. The percentage cumulative influence is relative to the cumulative footprint sum for each grid point.

(MSE) than STILT. Notably, the out-of-sample model performs on par with the in-sample model and slightly better than STILT, demonstrating successful generalization. Fluxes inferred using FootNet footprints are slightly more diffusive than those inferred using STILT footprints, though the difference in the distributions is small (see Figure S6).

Figure 8 shows the inversion results for the Barnett Shale. As in the Bay Area, FootNet produces smoother footprints than X-STILT while preserving the key spatial structure and magnitude of the posterior fluxes. The inferred methane emissions from both FootNet models are in close agreement with those from X-STILT. Evaluation against held-out TROPOMI observations shows comparable correlation and slightly improved MSE. Again, the out-of-sample FootNet performs similarly to the in-sample model and X-STILT.

Together, these results demonstrate that FootNet enables accurate, high-resolution flux inversions in regions entirely excluded from training. This eliminates the need for site-specific retraining or precomputed footprint libraries, establishing FootNet as a scalable solution for real-time greenhouse gas monitoring across broad spatial domains.

## 4 Interpreting model predictions: feature importance

To identify which inputs contribute most to model predictions, we applied the Permute-and-Predict (PaP; Fisher et al., 2019) method to 5,000 examples from the test set. The PaP method estimates the feature importance by measuring the drop in model performance when each input feature is randomly shuffled within a sample. A greater drop in performance indicates the higher importance of that feature to the model. This approach reveals the relative importance of each input to footprint prediction. Figure 9 shows the six most influential features for both surface and column models. In both cases, the Gaussian plume proxy is the top feature. This synthetic field encodes a directional prior based on surface winds and provides a strong initial guess of footprint location and spread. Zonal and meridional wind components at the time of observation also rank highly, consistent with their role in advecting plumes.

Interestingly, distance to receptor is also among the most important features. This variable captures spatial proximity and helps the model balance near and far-field sensitivity. Other meteorological features such as boundary layer height, temperature, and pressure appear less influential but still contribute meaningfully to the full prediction. The overall ranking of features is consistent across surface and column models, suggesting that both variants rely on similar transport-relevant signals. Supplementary Figures S2 and S3 provide a complete ranking of all inputs. Overall, the consistency of the identified features and their importance in atmospheric transport indicates that FootNet is learning physically meaningful drivers of atmospheric transport. Because FootNet is learning physically meaningful relationships, it is able to extrapolate to out-of-sample footprints with high fidelity.

## 5 Conclusions

Dense, high-resolution atmospheric GHG observations from surface networks and satellites offer the potential to constrain regional emissions at unprecedented spatiotemporal scales. However, the computational demands of physics-based atmospheric

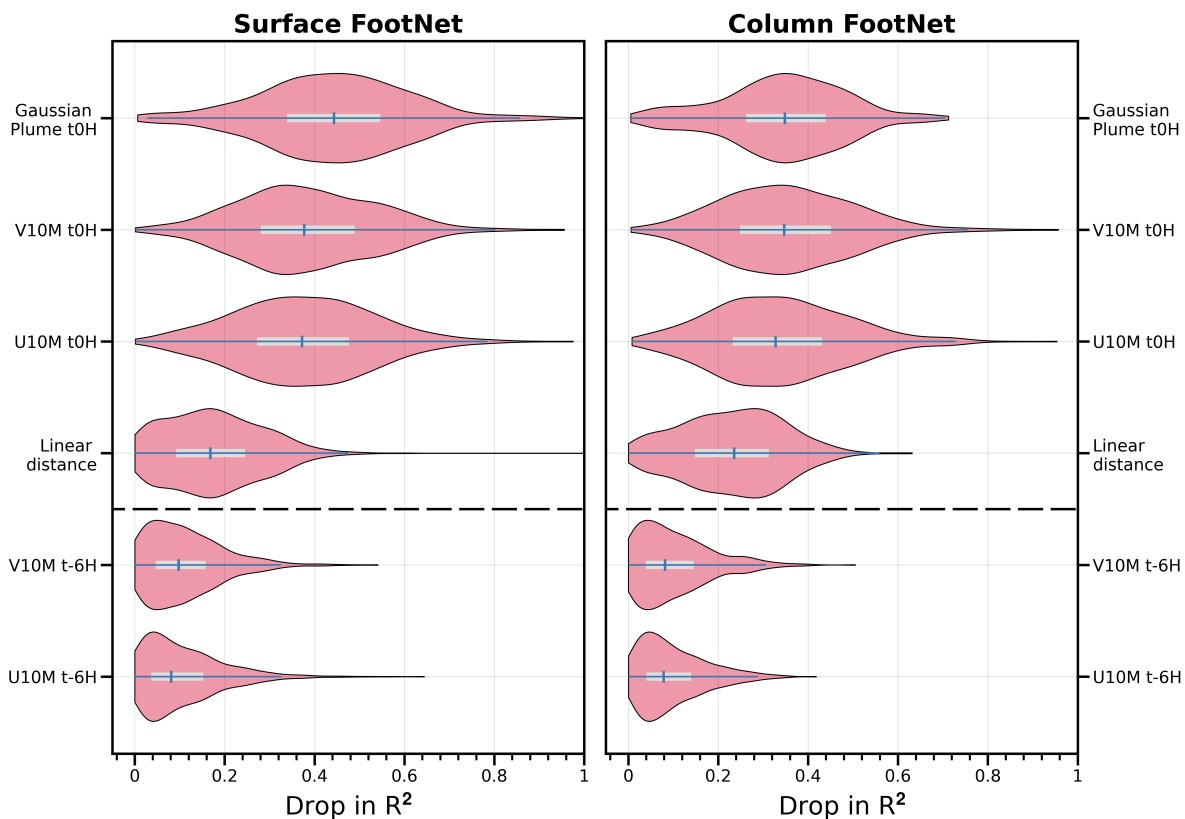

**Figure 9.** Top six input features ranked by importance using the Permute-and-Predict (PaP) method. Features are sorted by their relative impact on model loss when permuted. Surface and column models show consistent ranking patterns. Suffixes on the variable names indicate the timestep (e.g., "V10M t0H" means the v-component of the 10-meter winds at the receptor time whereas "V10M t-6H" is from 6 hours before the receptor time).

transport models have become a central bottleneck for flux inversion systems operating at these resolutions. Previous work introduced FootNet as a proof-of-concept deep learning emulator of atmospheric transport (He et al., 2025; Dadheech et al., 2025). In this study, we developed and evaluated FootNet v3, a machine learning emulator of atmospheric transport trained on half a million footprint examples across the contiguous United States. FootNet v3 uses a U-Net++ architecture and includes a soft mass-conservation constraint, enabling it to predict source-receptor relationships for both surface and column-averaged observations. It generalizes to previously unseen regions and meteorological conditions and is several orders of magnitude faster than traditional models, enabling it to function as a full surrogate for LPDMs without site-specific retraining or physics-based simulations.

We showed that FootNet replicates key structures in transport footprints across diverse terrain and weather conditions. Its predictions maintain skill even when driven with out-of-sample meteorological forcing. This was demonstrated by FootNet

accurately simulating footprints using GFS meteorology despite being trained with HRRR meteorology. We argue that FootNet is learning the fundamental relationship between meteorology and source-receptor relationships.

Further, we conducted GHG flux inversions with hourly $CO_2$ observations in the San Francisco Bay Area, and daily column-averaged methane observations in the Barnett Shale region. We conducted three GHG flux inversions for both of these regions: first with physics-based LPDM footprints, the second with in-sample FootNet footprints, and the third with out-of-sample FootNet footprints. In the Bay Area, both in-sample and out-of-sample FootNet outperformed STILT when evaluated against withheld $CO_2$ observations. In the Barnett Shale, FootNet performed comparably to X-STILT. These results demonstrate that FootNet v3 generalizes robustly across the CONUS region and does not require retraining to support high-quality flux inversions. These results strongly suggest that the FootNet model is robustly trained and generalizes well across the entire CONUS region and beyond. The consistency of the identified features and their importance in atmospheric transport indicates that FootNet is learning physically meaningful drivers of atmospheric transport and, as such, it is able to extrapolate to out-of-sample footprints with high fidelity. These findings show that machine learning models can learn the underlying physical relationships governing atmospheric transport, which allows them to extrapolate to out-of-sample scenarios. This framework used to develop the FootNet model can conceivably be used to emulate larger continental-scale transport. However, it may require other input features to better represent large-scale processes.

FootNet's ability to compute footprints in near-real time opens the door to scalable, low-latency GHG inversion systems capable of ingesting large volumes of in situ and remote sensing data. This work overcomes a critical computational bottleneck and paves the way for widespread deployment of flux inversion frameworks to support timely, actionable GHG monitoring across sectors and regions. Future extensions may focus on improving model interpretability, supporting probabilistic footprint estimates, and expanding training to include global domains. Nonetheless, the present results demonstrate that machine learning emulators can enable high fidelity footprints on-the-fly within a flux inversion. These emulators can further meet the accuracy and generalization demands of operational inversion systems and, for the first time, achieve out-of-sample transport fidelity sufficient for scientific and policy applications.

*Code availability.* The code for this study is available at https://github.com/nd349/FootNet and https://doi.org/10.5281/zenodo.16010441 (Dadheech and Turner, 2025a, last access: 16 July 2025). The basic tutorials on how to use the FootNet models are available at this website: https://footnet-uw.github.io/index.html.

*Data availability.* $CO_2$ data are available at http://beacon.berkeley.edu/Sites.aspx (Shusterman et al., 2016). TROPOMI methane data are available at https://dataspace.copernicus.eu/ (Veefkind et al., 2012). NOAA HRRR data is available at https://rapidrefresh.noaa.gov/hrrr/. 50 GB of example training was uploaded to https://doi.org/10.5281/zenodo.16011454 (Dadheech and Turner, 2025b, last access: 16 July 2025).

## Appendix A: Description of flux inversions

We performed a high-resolution hourly surface $CO_2$ flux inversion over the San Francisco Bay Area using hourly BEACO$_2$N $CO_2$ data from February 2 to May 5, 2020, including a 36 hours buffer on either end of the analysis window. We used a Bayesian framework across a $1\times1$ km$^2$ grid for this $CO_2$ urban flux inversion. The prior emissions were adapted from previous studies (McDonald et al., 2014; Turner et al., 2016, 2020; Dadheech et al., 2025). The state vector consisted of 15.4M elements, representing hourly average fluxes across space and time. We used the Kronecker product decomposition of spatiotemporal covariance to construct the prior error covariance matrix $\mathbf{B}$ (Yadav and Michalak, 2013). We assumed a 50% relative error for each state vector element with correlation lengths of 5 hours, one day, and 5 km. The measurement error, background concentration & error are adapted from Turner et al. (2020). The correlation lengths of 1 hour and 2 km were used for the off-diagonal terms of the observational error covariance matrix.

A high-resolution, column-averaged methane flux inversion was performed over the Barnett Shale using daily TROPOMI retrievals from February 1 to April 30, 2020, including a 7-day buffer on either end of the analysis window. The inversion was conducted using a Bayesian framework to optimize methane fluxes across a $1\times1$ km$^2$ grid. The prior emissions were based on the 2018 EPA gridded anthropogenic methane inventory (Maasakkers et al., 2023), downscaled from native resolution to 1 km using spatial disaggregation. These included emissions from fossil fuel production, waste, and agriculture. Biogenic or natural sources were excluded. The state vector consisted of 7.4 million elements, representing daily average fluxes across space and time. The prior error covariance matrix $\mathbf{B}$ was constructed as a separable spatiotemporal covariance using a Kronecker product decomposition (Yadav and Michalak, 2013), with 50% relative error applied to each state vector element and 7-day temporal and 5 km spatial correlation lengths. This setup yields a smooth and physically plausible prior while remaining computationally tractable. We used TROPOMI methane mixing ratio precision data as measurement uncertainties, and 7 days & 50 km of correlation lengths for the off-diagonal terms of the observational error covariance matrix. For simplicity, we assumed the model error is equal to the measurement error. The inversion domain includes a buffer to reduce the impact of the boundary conditions. To estimate the background column concentration and uncertainties for each observation, we implemented a directional sectoring scheme:

- The inversion domain was surrounded by eight azimuthal sectors.

- For each TROPOMI observation, the sector aligned with the prevailing wind direction was selected based on HRRR-derived wind fields.

- The mean methane concentration from TROPOMI pixels in that upwind sector (outside the inversion domain) was used as the background.

- The standard deviation of methane concentration from TROPOMI pixels in that upwind sector (outside the inversion domain) was used as the background uncertainty.

Wind direction and trajectory length were determined using a simple back-trajectory estimate based on particle travel time at the mean wind speed. Each TROPOMI ground pixel was divided into $1\times1$ km$^2$ subpixels to account for its spatial footprint.

For each subpixel, we computed source-receptor relationships (footprints) using one of three models: X-STILT (baseline),
in-sample FootNet, out-of-sample FootNet (see Figure 2). The subpixel-level footprints were then aggregated to the native
TROPOMI pixel resolution. Figure A1 shows this process. All three inversions used identical observational and prior configu-
rations to isolate differences due to the transport model.

## Appendix B:  Gaussian plume model

The Gaussian plume model is adapted from Nassar et al. (2017), and below are the equations we used to compute the Gaussian
plume:

$$V(x,y) = \frac{F}{\sqrt{2\pi}\sigma_y(x)u} e^{\frac{-1}{2}(\frac{y}{\sigma_y(x)})^2} \tag{B1}$$

$$\sigma_y(x) = a(\frac{x}{x_o})^{0.894} \tag{B2}$$

where $V$ is the vertical column in g/m$^2$ at and upwind of the receptor. The $x$ direction is parallel to the reversed wind
direction, and the $y$ direction is perpendicular to the wind direction. $F$ is the emission rate in g/s, which can be assumed as
a constant (here we assumed $F = 1$). $u$ is the wind speed, $\sigma_y$ is the standard deviation in the $y$ direction. $x_o = 1000$m is a
characteristic length, and $a$ is the atmospheric stability parameter, which we determine by classifying a source environment by
the Pasquill-Gifford stability (Pasquill, 1961; Nassar et al., 2017).

*Author contributions.* ND and AJT designed the research study. ND trained the models and conducted the flux inversions. ND and AJT
analyzed the results and wrote the manuscript.

*Competing interests.* The contact author has declared that none of the authors has any competing interests

*Acknowledgements.* This work originated from research supported by the National Aeronautics and Space Administration (grant nos.
80NSSC22K1557 and 80NSSC21K1808). It has continued as part of the FETCH$_4$ project and is supported by Schmidt Sciences through the
VESRI program.

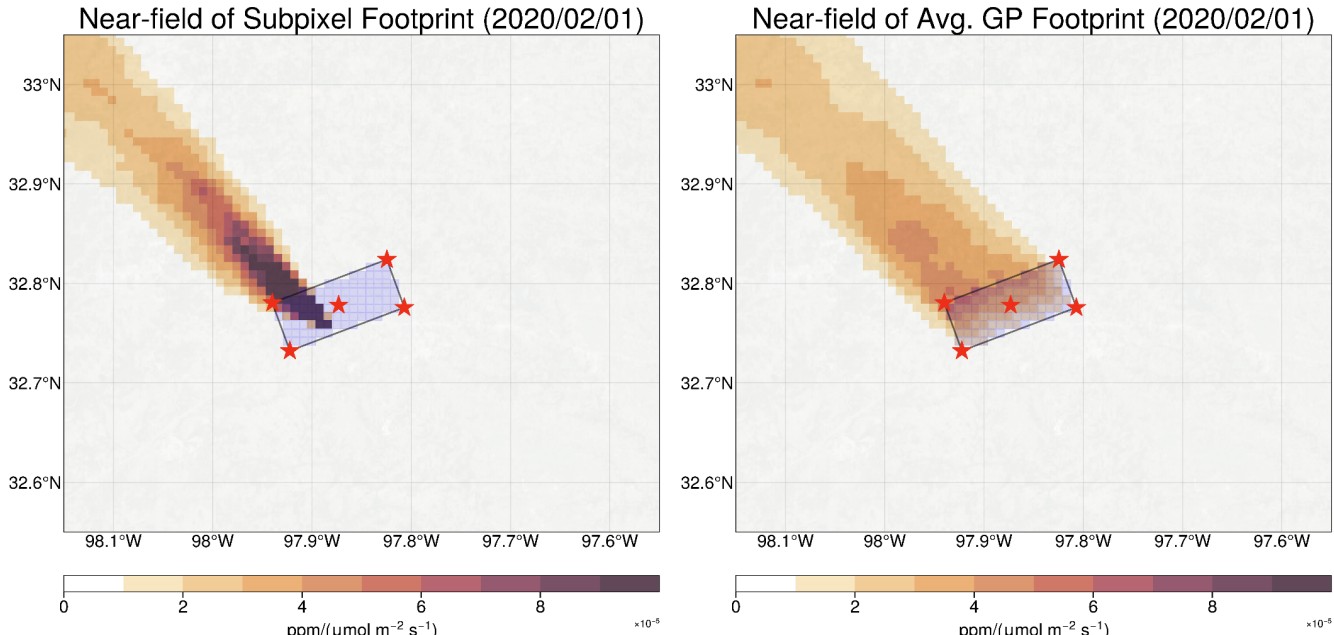

**Figure A1.** Construction of the TROPOMI footprint at $1 \times 1$ km$^2$ spatial resolution. Red stars indicate the bounding box and center of the TROPOMI pixel. Light blue boxes are $1 \times 1$ km$^2$ grid cells within the TROPOMI bounding box. (Left panel) shows a X-STILT footprint for a single $1 \times 1$ km$^2$ subpixel within the TROPOMI bounding box. (Right panel) shows the average footprint for all subpixels within the TROPOMI bounding box. The average footprint gives a lower maximum sensitivity but distributes it over a wide region. We also observe a gradient across the TROPOMI bounding box with increased sensitivity on the upwind (Northern) edge.

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
