# Peer review of "Simulating out-of-sample atmospheric transport to enable flux inversions"

_EGUsphere, 2025_

## Author Comment (AC1)

We thank the reviewers for their time and constructive comments on our manuscript. Our responses are color-coded in blue.

**Reviewer 1**

General Comments The paper addresses the emulation of footprints generated by atmospheric Lagrangian Particle Dispersion Models (LPDMs), which is an important problem in the field of trace gas inverse modelling, where the computational demands are increasing due to growing dataset sizes. In this paper, the authors demonstrate the performance of a new architecture for their Footnet algorithm (based on U-Net++). The model is evaluated in applications of inverse modelling of CO2 and CH4 based on in situ and column data. These evaluations are conducted in regions where the model has been trained as well as in "out-of-sample" regions. Therefore, the main novelty of the work lies in the model's ability to generalize to different meteorological conditions and geographic locations. Overall, I think the paper tackles an important subject that is within scope for ACP. It is generally well written and structured. However, before publication, I think so some elements of the work need to be explored more thoroughly, as there is a danger that the claimed generalizability is over-stated, particularly since the main claims are around the performance of the model in "out-of-sample" regions. In particular:

1. I think it is misleading to claim that this data-driven model out-performs a physics-based LPDM. The authors base this claim on an apparent improvement in fit to the mole fraction data when comparing Footnet or STILT to observations, attributing an improvement mainly to a tendency for the machine learning (ML) model to "smooth" the footprints (L150-L156). The way that the model has been trained (i.e., penalising any deviations from STILT footprints) means that perfect performance of the algorithm would be the exact retrieval of STILT footprints. If the ML model does not fit the STILT output perfectly (which, of course, it can never do), but better fits the independent observations, this improvement must be coincidental. Put another way, any difference in performance to STILT must be considered a degradation in emulator performance (even if it's better fit to some other observational dataset). What the authors have found here is potentially interesting, and it would imply that we should smooth LPDM outputs to improve the fit to the data. But, to me, this points to a separate systematic model error or representation issue, rather than some benefit that somehow comes from training an emulator.

We agree that FootNet, being trained to reproduce STILT footprints, cannot exceed STILT's physical fidelity. The ultimate goal is to simulate atmospheric observations and infer fluxes from atmospheric observations. What we have shown is that FootNet outperforms STILT in this objective. As we describe in the text, the smoother output can reduce meteorological bias/noise sensitivity and yield more stable inversion results when evaluated against independent GHG observations. The improved fit to mole fraction data should therefore be interpreted as a *practical* advantage stemming from reduced representation errors and smoother source-receptor fields, not as a fundamental improvement in atmospheric transport modeling. We have added/updated text:

Line 157: However, it is important to note that FootNet is trained on STILT and X-STILT. As such, the improved performance against independent atmospheric observations implies that the smoothness is mitigating underlying model errors.

2. To me, it seems that the claim of "out-of-sample" generalization has only been partially demonstrated, since the tests in "unseen regions" were performed in 2020, which is the time period during which the inversion was being trained. Therefore, the model has "seen" similar footprints around the same time, within a few hundred km of the left-out region. Meteorological variables have substantial spatial and temporal correlations. To be more out-of-sample and strengthen the claims of the work, it would be beneficial to demonstrate the model performance in these regions in a different year (e.g., 2022)? Additionally, it could also be tested in another country, but I accept that this would be more challenging to set up.

We thank the reviewer for this helpful comment. In response, we removed the 2020 data from the training set and retrained the out-of-sample FootNet model using only 2021 data, while keeping the spatial domains consistent with the previous configuration. The updated inversion results are similar to those obtained previously, supporting the generalization capability of FootNet. We have also revised the manuscript text accordingly and updated Figures 7 and 8.

Line 83: These 500,000 footprints were split into two training configurations. The first, which we refer to as the *out-of-sample FootNet*, excludes all data from 2020 and two regions: the bulk of California and much of Texas. The out-of-sample FootNet uses footprints from 2021 for training. Within these two regions are two case studies reserved for later out-of-sample evaluation: the San Francisco Bay Area (Domain A) and the Barnett Shale region (Domain B). These out-of-sample evaluations are done using the observations from 2020.

Figure 7

Figure 8

3. The authors claim that the model has also been demonstrated against out-of-sample meteorology, but I think this is probably over-stating what has been achieved. The two products (GFS and HRRR) are based on assimilated meteorological observations, and, whilst I don't have direct experience of these products, I'm assuming that they must be extremely similar to each other, especially in the lower troposphere. Therefore, can the GFS dataset really be considered "out-of-sample"? It will surely be almost fully correlated with HRRR.

We thank the reviewer for this important clarification and agree that HRRR and GFS will have strong similarities, as both assimilate meteorological observations. However, many important parameters for LPDM models are not assimilated (e.g., PBL height). Further, the spatial resolution of these meteorological models differs by more than a factor of 3. Specifically, HRRR operates at 3 km resolution and is limited to CONUS, whereas GFS has a coarser (~10 km) global configuration. Our intent was not to imply complete statistical independence between these products, but rather to test FootNet's robustness to different meteorological forcings with distinct resolutions and model physics. The experiment therefore demonstrates that FootNet maintains skill when applied to a coarser, globally available meteorology product, an important step toward extending FootNet's applicability beyond CONUS. We have also added this brief discussion to the manuscript.

Line 184: This is important because HRRR is only available over CONUS, whereas GFS is a global product. This means that FootNet can be used in domains outside of CONUS with GFS or any other global meteorology products.

4. The ML model performance has been demonstrated over relatively small scales (~400km x 400km). At these scales, I'm assuming the inversions are strongly influenced by potential errors in boundary conditions. I don't think this is a problem with the approach per se, but it does seem like a limitation that should be mentioned, since it may limit the extension of the model to some regions.

The output of the FootNet model (the footprint) is independent of the boundary conditions for the inversion. The FootNet domain (400km x 400km) is chosen to be substantially larger than the domain typically used for urban flux inversions to reduce the impact of the boundary conditions. However, the reviewer is correct that all regional flux inversions are dependent on boundary conditions. This is true for inversions using physics-based transport models and ML models.

In our work, we use identical boundary conditions for the inversions using STILT and FootNet, therefore differences in the results are not due to boundary conditions. Appendix A discusses the method we used for computing boundary conditions.

Line 311: The inversion domain includes a buffer to reduce the impact of the boundary conditions.

5. The claim that the model has "learned the physics" governing atmospheric transport also seems like it could be misconstrued (L14, L64 and elsewhere). I suspect that some readers will assume this is an application of physics-informed machine learning, where the model has been informed by some underlying physical equations. In this case, it's a purely data-driven approach, so perhaps it's better described as a having learned relationships between meteorology and LPDM footprints.

We have updated instances in the text to from "learned the physics" to "learned the relationship".

**Other general comments:**

6. No details are provided on the testing/validation set or metrics, which is critical for the evaluation of this kind of paper. This certainly needs addressing, with justification provided for the test/train split and choice of metrics. L140 and Figure 5 seems to imply that the testing set is from the same year and location(s) as the training set. Given the above-mentioned strong spatial and temporal correlations in the atmosphere, this seems to be a critical limitation. It seems imperative that the testing set is at least temporally distinct (i.e., separated in time by more than synoptic timescales) from the training set (and, if the authors want to demonstrate spatially out-of-sample performance, spatially distinct too).

Thank you for the suggestion. We have now added the text on the train test split.

Line 88: For validation and testing, we randomly picked multiple 400x400 km2 subdomains (similar to Figure 2) with enhanced temporal sampling, as well as footprints computed for randomly sampled receptors across CONUS that do not correspond to the training receptor locations. We used 50,000 footprints randomly sampled in 2020 and 2021 for testing and validation.

**Specific comments:**

7. Throughout, the authors use "observations" to describe the training footprints (L5 and elsewhere). I found this confusing, as, to me, this would imply mole fraction data. Why not say that, e.g., "500,000 training footprints"?

Thank you for the suggestion. We have now replaced pseudo-observations with receptors and footprints throughout the manuscript.

8. Are the results for one random seed only or is it a mean over several seeds? If it is a mean over several seeds, could some measure of error or standard deviation be shown on Figure 3?

The results are for one random seed only. Training both Surface and Column FootNet models with several seeds can quickly become expensive, as it can take more than a week to finish a single training, as the training data exceeds 100,000 examples.

Figure Caption 3: Results shown are for a single random seed.

9. In Figure 1, there are different circles for the arrows and the different convolutions, what do the colors represent?

We have updated the text.

Figure Caption 1: The green circles and green & blue arrows represent new layers and their respective connections compared to the earlier U-Net architecture.

10. Line 23: There is another preprint under consideration for GMD that covers very similar themes using a different modelling approach, Fillola et al. (2025): https://egusphere.copernicus.org/preprints/2025/egusphere-2025-2392/. It seems that these two papers should cite each other in their revised forms.

We thank the reviewer for this suggestion. We have now cited this manuscript.

Line 21: However, the computational burden and data storage demands of physics-based atmospheric transport models limit the scalability of current inversion frameworks, especially when leveraging dense GHG observations (Roten et al., 2021; Cartwright et al., 2023; Fillola et al., 2023, 2025).

11. Line 133, is there any hypothesis on why there was no clear benefit to including a mass penalty?

We investigated but, unfortunately, were unable to draw a meaningful conclusion as to why and prefer not to speculate here.

12. Line 148 – 149: As mentioned above, I can't see how the logic here holds up. Unless you're bringing in some additional information, no matter what the biases in the model you're emulating, the best you can do is emulate that model, biases and all.

We thank the reviewer for raising this important point. We point to the response to comment 1 and our updated text below:

Line 157: However, it is important to note that FootNet is trained on STILT and X-STILT. As such, the improved performance against independent atmospheric observations implies that the smoothness is mitigating underlying model errors.

13. Line 150: What do you propose is the reason for the smoothness?

Machine learning models generalize by identifying an underlying trend instead of fitting every noisy data point. Additionally, FootNet models are built on U-Net and U-Net++ architectures, which consist of Convolutional Neural Network (CNN) layers. These layers contain convolution kernels that process the data by performing convolution operations on local neighbourhoods instead of individual data points. We have now added this text to the manuscript.

Line 165: Machine learning models generalize by finding an underlying trend instead of fitting every noisy data point (Shukla et al., 2021). FootNet model architecture consists of Convolutional Neural Network (CNN) layers that perform convolution operations on local neighborhoods instead of processing individual pixels (Bishop, 2006). These are the two primary reasons why the FootNet model produces smoother outputs.

14. Line 164, to make these claims, you'll need to explain what is substantively different about these two meteorological products and demonstrate that the meteorological variables are substantially different (see general comment above)

We thank the reviewer for this comment. As discussed in the response to comment 3, we agree with the reviewer that the HRRR and GFS meteorologies may have similarities. However, as mentioned above, there are significant operational differences between them. While HRRR operates at 3km spatial resolution, GFS operates at 10km or greater spatial resolution. This analysis helped us investigate that FootNet continues to have skill in its prediction even when we use coarser meteorology to generate footprints at 1 km resolution.

Line 184: This is important because HRRR is only available over CONUS, whereas GFS is a global product. This means that FootNet can be used in domains outside of CONUS with GFS or any other global meteorology products.

15. For Figure 6, I understand that the results are similar for different types of meteorology but I do not understand the axes, since they seem to be displaying aggregated 2D quantities (footprints). Are these all of the gridded footprint values

from all locations aggregated together and compared? How is the r value computed in this case? Can you clarity this?

Each data point represents one of the non-zero influence pixels of randomly sampled 5000 footprints from the test dataset. The r value is computed using the vectors of all non-zero influence pixels for STILT and FootNet. We have now added this information in the figure 6 caption as well.

Figure Caption 6: Each data point in the plot represents a non-zero influence pixel from one of the 5000 randomly sampled footprints in the test set.

16. Figure 7 and 8, can you clarify what the percentage is relative to and what quantile the contour corresponds to?

We have updated the text to explain this:

Figure Captions 7 & 8: White contours in the left column represent the 60th percentile of the cumulative footprint influence. The percentage cumulative influence is relative to the cumulative footprint sum for each grid point.

17. Line 271: Presumably a standard analytical Gaussian inversion? Provide a few extra details or a reference. How were model and measurement uncertainties (and emulator uncertainties?) represented?

We thank the reviewer for this suggestion. We have now added the details on model and measurement uncertainties in the manuscript.

Line 309: We used TROPOMI methane mixing ratio precision data as measurement uncertainties, and 7 days & 50 km of correlation lengths for the off-diagonal terms of the observational error covariance matrix. For simplicity, we assumed the model error is equal to the measurement error.

**Reviewer 2**

This is a well-written manuscript of a nice study, and very well-organized. I especially appreciate that the authors extended the comparison analysis through performing flux inversions in order to understand the impact of using the FootNet footprints when deriving fluxes. I have only skimmed the previous two papers by this same group on this topic, but it seems that in this one they have extended the model to work on column (satellite) GHG data, and that they have evaluated/validated the results for out-of-sample data, which is probably the main contribution here. The editors can determine if perhaps the manuscript is better suited for GMD, as it is very much a model development & validation study. I recommend publication after addressing the comments below.

**Overall comments:**

18. In regional or city-scale inversions, particle trajectories from ATD models like STILT are also often used to sample and/or optimize a background in some way. Can this be done with Footnet?

Unfortunately, FootNet only yields the time-integrated footprint. However, we use input wind fields to compute the upwind region and boundary conditions. Appendix A discusses the computation of upwind boundary conditions.

19. In the out-of-sample FootNet simulations (i.e. when the footprint was generated using a model that did not use the Barnett region for training), were the simulations also from a different year or month than what was used in training? I.e., I am wondering how well Footnet performs for data (receptors) from a completely different year than the training. I would think that transport is correlated across very large spatial scales and it could be that if trained on data from the same time period, could allow the model to perform better, even if the receptor is hundreds of km away from the training data receptors? Now reading L 195, perhaps this was already done in previous work, perhaps that could be mentioned either way.

We thank the reviewer for this comment. We have now revised the out-of-sample FootNet training and evaluation. We now only use data from 2021 to train the out-of-sample FootNet model and test it on data from 2020. We found that the model performance does not change in both one-to-one comparisons of footprints and in the inversion.

We have updated the text and figures with the revised out-of-sample FootNet version. See response to comment 2.

20. Optional question for the authors to perhaps comment on in the Conclusion or future work: How can FootNet can be used for larger continental-scale inversions. Will v4 simulate longer time periods (at coarser scales) to perform inversions for CONUS?

We thank the reviewer for this question. The FootNet model can conceivably be trained on continental-scale transport with longer time periods and coarser spatial resolutions. This may require additional input features to rigorously resolve processes that dominate the transport at continental scales (e.g., convection). However, we do not plan to do this in the near-term. Logistically speaking, the lead author (Nikhil Dadheech) is now using FootNet for a science application as his final PhD chapter.

Line 273: This framework used to develop the FootNet model can conceivably be used to emulate larger continental-scale transport. However, it may require other input features to better represent large-scale processes.

21. Lastly, it seems the code is available. Would the authors recommend that others use the already-trained FootNet code to generate footprints for their own use? What would be the caveats about the use of this model (where would it perform well or not)?

Yes, we recommend others to use the already-trained FootNet as it has been trained over the entire CONUS, and its generalization has been evaluated in this work. The training code is also available for users who would like to fine-tune the FootNet for

any specific use. We have also developed a website and some basic tutorials. We have added the website to the updated manuscript:

Line 285: The basic tutorials on how to use the FootNet models are available at this website: https://footnet-uw.github.io/index.html.

**Abstract**

22. I would add "dispersion" to "atmospheric transport and dispersion models" here in the abstract at least, as dispersion is a large part of the modeling that Hysplit/Stilt is doing, along with transporting the tracer, and that is a common term in the literature (ATD models).

We thank the reviewer for this suggestion. We have now updated the abstract.

Line 1: Accurately estimating greenhouse gas (GHG) emissions from atmospheric observations requires resolving the upwind influence of measurements via atmospheric transport and dispersion models.

23. L47-48 adjust grammar in this sentence appropriately – what does "which" refer to (L48). Perhaps omit "the" prior to "flux inversion"?

We thank the reviewer for raising this grammatical mistake. We have now corrected it in the text.

Line 48: Variational methods such as 4D-var are also popular in flux inversions, which require an adjoint of the Eulerian model to compute the atmospheric transport (Henze et al. 2007)

24. L75+ Perhaps the authors could clarify what they mean by "pseudo-observation" - is this a simulated GHG concentration, or is it a footprint (i.e. gridded and varying in space and time)? Readying on in L86, it seems the observations were simulated with footprints but – I would think the observations are footprints themselves, right? The output after all is a footprint.

We have now replaced "pseudo-observations" with "footprints" and "receptors" throughout the manuscript.

25. L88 - why was HRRR re-gridded to 1-km? Using STILT does not require that even if the STILT grid is at 1-km... does FootNet perform better when this is done— in which case I would guess it depends how the interpolation is done?

Good question, this is something that was tested early on (i.e., He et al., GMD 2025). Our target was 1-km GHG flux inversions. Using 1-km input fields allowed flexibility in which fields were used. For example, it meant that we could use topography at 1-km resolution. Ultimately, we found better performance with 1km regridded inputs compared to native 3km input fields. The primary reason for the drop in performance is that the input fields will have different resolutions. We have now updated the text.

- Line 100: This is done to ensure that all the input fields (e.g., meteorology, Gaussian plume, etc.) are at the same spatial resolution.
- 26. L94- How far back were the particles traced in the training footprints? From this time step list, it seems it would only be 24 hours. How does this affect the simulated column footprints, especially the influence of the upper altitudes where the particles may not have any footprint influence in the first 24 hours at times?

For STILT and X-STILT, we simulated the trajectories for 72 hours or until the particles leave the domain. However, we only use 24 hours of meteorology before the measurement time as input fields in the FootNet. We conducted a series of tests using varying amounts of meteorological data including further than 24 hours back in time. We found minimal improvement in performance beyond 24 hours. Including additional meteorological data does increase the data loading time and, as such, inference time.

- Line 94: The trajectories were simulated backward in time for 72 hours, or until the particles exited the domain, whichever occurred first.
- Line 102: We found minimal improvement in performance when including meteorological inputs more than 24 hours backward in time.
- 27. L169 Is there a reference for how much uncertainty there is in the STILT footprints? Thinking about the comparison between FootNet and STILT in the context of the overall uncertainty in STILT may be a useful framing and can more quantitatively make this point. The papers looking at differences between different models probably only compare in certain places and times making extrapolation or generalization difficult, but one could cite some here as a comparison- is the uncertainty 10%, 20%, 50%? (e.g., Karion et al., 2019, https://doi.org/10.5194/acp-19-2561-2019 is in the Barnett so could be useful for making this point?).

We thank the reviewer for suggesting Karion et al. 2019 as a supporting citation. Munassar et al. (2023) observed STILT and FlexPart to have a relative difference of 61% in the flux inversions over Europe. We have now cited both of these studies in our manuscript.

- Line 187: For instance, STILT and FLEXPART can produce similar or larger disagreements than observed here when run under similar conditions (Karion et al., 2019; Munassar et al., 2023).
- 28. Fig 5 regarding the smoother footprints generated by FootNet: Were the STILT footprints run with the optional far-field smoothing (Gaussian Kernel method) that is provided with the University of Utah STILT footprinting codebase? If so, how was it set?

Yes, we used the default Gaussian Kernel smoothing parameters provided by the STILT codebase.

29. SI Fig S2 and Fig S3, perhaps note in the legend that GP refers to Gaussian Plume or define in the table in Fig 1 next to "Gaussian Plume (GP)", for example, for consistency.

We thank the reviewer for this suggestion. We have now added this to the figure captions.

30. Appendix A: It would be useful to include a similarly short description of the details of the SF inversion so the reader does not need to refer to the previous papers, if possible.

We have now added a description of the SF inversion in Appendix A.

31. Fig 6, units should be included - presumably these are summed over space and time for each receptor (so each data point in the color scale is a full footprint?).

We have now added the units in Figure 6. Each data point in the color scale is a non-zero pixel of a footprint from randomly sampled footprints from the test set. We have updated the figure caption to emphasize the same.

Figure Caption 6: Each data point in the plot represents a non-zero influence pixel from one of the 5000 randomly sampled footprints in the test set. The unit of the footprints is ppm/( $\mu$ mol m-2 s-1).

32. Fig 6A does indicate a high bias in the FootNet footprints - can the authors comment on this?

The high bias in the FootNet footprints corresponds to small magnitudes of the footprint, particularly in the far-field, where the smoothness of the footprint structure causes a difference from the ground truth footprints from STILT. We have now added this comment in the manuscript.

Line 178: Notably, higher bias occurs for small footprint values, particularly in the far-field, where the smoothness of FootNet footprints results in deviations from the ground truth STILT footprints.

33. Figs 7 & 8: Another figure or panel should indicate the difference between the posterior fluxes for each case (relative to the STILT or XSTILT case, plus comparing the in-sample vs. out-of-sample FootNet, either absolute units of percent perhaps). As is, the second column really looks identical. Especially in Fig 7, the footprints look quite different between panel A vs. D and G, so it would be useful to see the magnitude of the flux difference.

We thank the reviewer for this suggestion. We have now added flux difference plots (Figure S4 & S5) in the supplement. The magnitude difference is observed to be less than 6% for both the SF Bay Area and Barnett inversions.

34. L215: given the importance of the Gaussian plume proxy, can the authors include the basics of how this was calculated (equation?) – perhaps in the SI or appendix. For example, how was the stability class determined for each case? It is indeed interesting that this is the most important input, showing that giving the model some basic understanding of the relationship between the inputs helps it perform better, rather than giving the model only the inputs to the GP proxy, for example. Perhaps this points to the model not actually "learning" relationships, since the GP gives it the basic expected relationship between wind, PBL, etc.

We have added an appendix section (Appendix B) discussing the Gaussian plume model we used (Nassar et al. 2017). Here are the equations we used to compute Gaussian plume:

$$V(x, y) = \frac{F}{\sqrt{2\Pi}\sigma_{y}(x)u}e^{\frac{-1}{2}(\frac{y}{\sigma_{y}(x)})^{2}}$$

$$\sigma_{y}(x) = a(\frac{x}{x})^{0.894}$$

Where V is the vertical column in  $g/m^2$  at and upwind of the receptor. The x direction is parallel to the reversed wind direction, and the y direction is perpendicular to the wind direction. F is the emission rate in g/s, which can be assumed as a constant (here we assumed F=1g/s), u is the wind speed,  $\sigma_y$  is the standard deviation in the y direction.  $x_o = 1000m$  is the characteristic length, and a is the atmospheric stability parameter which we determine by classifying a source environment by the Pasquill-Gifford stability.

**Reviewer 3**

**General Comments**

This paper introduces FootNet v3. FootNet is a deep learning emulator of atmospheric transport that computes the sensitivity of passive atmospheric trace gas concentrations to upwind emissions (the "footprint"). The footprint is a key component of inverse analysis for emissions quantification, but is a computational bottleneck that limits the feasibility of the analysis. FootNet promises to provide footprints with 650x less wall clock time per footprint vs Lagrangian Particle Dispersion models or Eulerian transport models, while maintaining or even improving model fidelity. FootNet v3 specifically promises to provide footprints for locations and times outside the training data, allowing the model to be deployed to new column and point concentration observations without re-training. This would represent a major step forward in the field of inverse analysis for emissions quantification by greatly decreasing cost and expanding access to more researchers.

FootNet v3 improves upon previous versions of FootNet by: Using the improved U-Net++ architecture in place of the U-Net architecture of previous FootNet versions. Increasing the quantity of training footprints to 500,000 and increasing the breadth to cover the entire Continental United States across seasons and meteorological conditions. These training footprints were computed using XSTILT driven by HRRR meteorology with a The paper makes the following key claim about FootNet v3 that must be justified:

FootNet v3 can produce footprints at locations and times outside the training data, towards generalization to new observations.

To justify this claim, the authors train and run FootNet v3 in a "out-of-sample" configuration, where training data from large regions in California and Texas are withheld and the resulting model is used to compute flux inversions using real observations of 1) surface CO2 measurements from the BEACO2N network in San Francisco, and 2) TROPOMI XCH4 observations in the Barnett Shale of Texas. The authors find that FootNet v3 performs comparably and even slightly better than STILT alone in the out-of sample inversions, demonstrating that FootNet v3 could indeed be used for regions out of the training sample. These experiments appear to be well done.

Additional claims include that the quantity of samples and sample strategy were sufficient to constrain the model, which was demonstrated by a validation loss experiment, and that the model appropriately conserved mass, which was tuned with a parameter for surface footprints.

This paper is well written and technical comments about the writing are minor. I support publication of this paper with minor revisions.

35. One question I do have: The authors demonstrate here and in their previous papers that the added diffusivity of FootNet enhances some properties of the predicted concentrations, but how does if affect the distributions of retrieved emitters? Does it induce more diffuse posterior emissions patterns? What are the implications for modeling emissions from the "fat tail" distribution of methane emissions from point sources? This is one of the key questions that could be answered by high resolution satellite data, which would benefit most from having such a computationally efficient transport model.

We thank the reviewer for this insightful comment highlighting an important implication of FootNet's enhanced diffusivity. We agree that FootNet's smoother footprints could influence the spatial distribution of posterior emissions, potentially leading to more diffuse flux estimates compared to those derived from highly localized LPDM footprints. We attempt to quantify this by looking at cumulative distribution function of the fluxes from STILT inversions and FootNet inversions for the SF Bay Area. This is now included as Supplemental Figure S6 and shown below. Fluxes inferred using FootNet footprints appear *slightly* more diffusive than those inferred using STILT footprints, though the difference is minor.

Regarding the latter point on the implications for the "fat tail" of methane emissions, this is something we plan to investigate in an inversion using TROPOMI satellite observations for Dadheech's final PhD chapter.

Figure S6: Cumulative Distribution Functions (CDFs) of posterior and prior differences for STILT and FootNet footprints in the San Francisco Bay Area.

Line 221: Fluxes inferred using FootNet footprints are slightly more diffusive than those inferred using STILT footprints, though the difference in the distributions is small (see Figure S6).

**Specific Comments**

36. How much influence in the out-of-sample footprints lies in the in-sample domain? If this is significant it could taint the results. A stronger experiment would be to fully separate the out of sample domain in time and space, though I suspect the spatiotemporal domain was chosen to align with previous work to conserve limited resources, which is understandable.

We thank the reviewer for this comment. First, we have revised the out-of-sample model to train it solely on data from 2021 and test it on 2020. Second, the footprints computed for the inversions have zero influence in the spatial domain in which the out-of-sample FootNet model was trained. This ensures full separation of the spatiotemporal domain for the out-of-sample tests.

Line 83: These 500,000 footprints were split into two training configurations. The first, which we refer to as the *out-of-sample FootNet*, excludes all data from 2020 and two regions: the bulk of California and much of Texas. The out-of-sample FootNet uses footprints from 2021 for training. Within these two regions are two case

studies reserved for later out-of-sample evaluation: the San Francisco Bay Area (Domain A) and the Barnett Shale region (Domain B). These out-of-sample evaluations are done using the observations from 2020.

37. Line 116 "but also suggest that moderately sized, region-specific training efforts may be sufficient to fine-tune for local applications" It is not obvious to me that this point follows from the data provided.

We thank the reviewer for this helpful observation. We agree with the reviewer that this statement can be confusing. We have now removed this line from the manuscript.

38. Line 147: "While STILT's spatial structure is more physically realistic (as it directly solves the governing equations), it is not necessarily more accurate due to potential biases in the driving meteorological fields. Highly localized but biased footprints could introduce artifacts in GHG flux inversions. In this context, the smoother prediction from FootNet may actually be preferable." STILT has a feature that can artificaially increase the dispersion and induces similar smoothness. See Lin and Gerbig (2005) Accounting for the effect of transport errors on tracer inverrsions, Geophys Res Lett 32 L01802 doi:10.1029/2004GL021127. Also, correlated errors along the duration of a STILT footprint are implemented in Jones et al. (2021) Assessing urban methane emissions using column-observing portable Fourier transform infrared (FTIR) spectrometers and a novel Bayesian inversion framework Atmos. Chem. Phys., 21, 13131–13147, <a href="https://doi.org/10.5194/acp-21-13131-2021">https://doi.org/10.5194/acp-21-13131-2021</a>.

We thank the reviewer for pointing these papers out. We have updated the text to mention additional approaches to mitigating transport errors.

Line 159: We also note that other approaches have been developed to mitigate transport errors within physics-based models and inversion frameworks. For example, incorporating stochastic wind uncertainties to enhance dispersion (Lin and Gerbig, 2005) and explicitly accounting for correlated transport errors over the footprint duration (Jones et al., 2021).

39. Line 167 "This means that FootNet could likely be used in domains outside of where it was trained." I thought that this was tested directly by excluding footprints in this domain from the training set. I think that the conclusions that can be drawn from the HRRR vs GFS comparison are analogous to those that would be drawn from alternate met products used in traditional observation system simulation experiments.

We agree that the HRRR-GFS comparison is analogous to using alternate meteorological forcings in traditional observation system simulation experiments. Our intent here was to emphasize that although FootNet was trained using HRRR meteorology at 3km native resolution, it retains skill when applied to GFS meteorology, which has a coarser (~10 km) global resolution. This result is important because HRRR is limited to CONUS, whereas GFS is available globally. Therefore, this analysis demonstrates that FootNet can be applied with GFS to regions outside

CONUS where HRRR data are unavailable. We have revised the text to clarify this point.

Line 184: This is important because HRRR is only available over CONUS, whereas GFS is a global product. This means that FootNet can be used in domains outside of CONUS with GFS or any other global meteorology products.

**Technical Corrections**

40. In He et al., 2025, the FootNet version is named FootNet v1.0. In Dadheech et al 2025 the version number is omitted. Should this not be FootNet v2.0? What is the minor version number signifying in He et al., 2025?

The minor version number signifying in "He et al., 2025" is a requirement of GMD. Our original submission of the He et al. paper was "FootNet v1". We were required by the journal to change it to v1.0. The restrictions imposed by GMD are part of why we did not submit later manuscripts there. The Dadheech paper referred to the model used there as FootNet v2 because there were a few important changes needed to enable the model to be used in flux inversions

Regarding versioning, these are the versions:

- FootNet v1 He et al. GMD (2025): The paper was originally submitted in 2023 to GRL. It was ultimately published in 2025 in GMD.
- FootNet v2 Dadheech et al. ACP (2025): This paper was submitted in 2024 and published in 2025.
- FootNet v3 this manuscript
- 41. In this paper, Eulerian transport models and Lagrangian Particle Dispersion models are referred to as "full-physics". These models rely heavily on parameterizations to simulate dispersion, which is a fundamental property of the output, and therefore full-physics is not an appropriate term.

We thank the reviewer for this suggestion. We have now replaced "full-physics" with "physics-based" throughout the manuscript.

42. The descriptions of the domains in the caption of figure 2 is confusing and I think an error— It says that the gray dots indicate individual pseudo observations, and that the Domains A and B are withheld from the out-of-sample training configuration, but there is a wide margin of missing gray dots around Domains A and B. Should the full set of pseudo-observations cover all of CONUS (as is given in the supplement) and the out-of-sample set be the dots drawn? This would better align with the description in the text. Also, this caption refers generically to FootNet when it should refer to FootNet v3.

The reviewer is correct, we have updated the caption to better explain this and refer to the model as FootNet v3.

Figure Caption 2: (Top panel) Receptors for training out-of-sample FootNet. Gray dots indicate individual receptors sampled uniformly across CONUS. Red stars show centers of 400x400 km² subdomains with enhanced temporal sampling. Domains A and B are withheld in the out-of-sample training configuration. Supplemental Figure S1 shows the full training data used for the in-sample FootNet model.

43. Line 96 "Table in Figure 1" -> "The 'Inputs' table in Figure 1"?

We thank the reviewer for this suggestion. We have now revised the text.

44. Figure 4 caption: "the surface model" should be replaced with "Surface FootNet" so that the figure stands alone.

We thank the reviewer for this suggestion. We have now revised the Figure 4 caption.

Figure Caption 4: Trade-off between validation loss (mean squared error; MSE) and percentage footprint mass difference for different values of  $\alpha$  in the surface FootNet model.

45. Line 169: "For instance, STILT and FLEXPART can produce larger disagreement than observed here when run under similar conditions. This flexibility allows FootNet to support flux inversions using multiple sources of meteorology" citation required.

We have now added the citations to this.

Line 187: For instance, STILT and FLEXPART can produce similar or larger disagreements than observed here when run under similar conditions (Karion et al., 2019; Munassar et al., 2023).

46. Figure 6: What is the property being evaluated? Sum of footprint weight?

The property being evaluated is the non-zero influence pixels of randomly sampled 5000 footprints from the test set. We have now revised the Figure 6 caption.

Figure Caption 6: Each data point in the plot represents a non-zero influence pixel from one of the 5000 randomly sampled footprints in the test set.

---

## Author Response (AR2)

We thank the reviewers for their comments. Our responses are color-coded in blue.

We thank the authors for their detailed rebuttal and appreciate the substantial work undertaken to retrain the model and revise the manuscript. The revisions improve the clarity of the contribution. However, several issues remain that should be addressed before publication to ensure the claims are appropriately supported and not overstated.

Major points
1. Claims of FootNet outperforming STILT remain too strong.
Although the authors acknowledge in the rebuttal that FootNet cannot exceed the physical fidelity of STILT, the abstract still states that the emulator "out-performs STILT." This wording remains misleading because FootNet is trained explicitly to reproduce STILT outputs. Any differences relative to STILT are, by definition, deviations from the training target. If these deviations lead to a better match against independent observations, that is interesting but should not be framed as FootNet outperforming STILT. I strongly suggest softening the language in the abstract and throughout as agreed in the rebuttal.

This is an important point and we disagree with the reviewer here. Our final goal is to conduct GHG flux inversions and our primary metric of success is observations that indicate a successful GHG flux inversion. Our work demonstrates that FootNet matches or exceeds the performance of physics-based models **when used in a flux inversion** and was evaluated against independent GHG observations (see Figures 7 and 8). Again, our flux inversion comparisons are with respect to real independent observations in the atmosphere and, as such, FootNet **can** outperform STILT in such comparisons. We have updated the text in the manuscript to reflect this.

Line 10: Case studies using GHG measurements in the San Francisco Bay Area and Barnett Shale show that FootNet matches or exceeds the performance of physics-based models when used in a flux inversion and evaluated against independent GHG observations.

2. Clarification and consistency on the testing/validation datasets.
It is good to see that the authors have removed 2020 from the training dataset so that tests can be performed on unseen periods during 2020. However, this separation of the training and testing/validation datasets should be enforced throughout the paper. Furthermore, the distinction between validation and test datasets is still unclear. As written, they appear to be the same set?
The rebuttal states that the testing and validation set includes randomly sampled footprints from 2020 and 2021. If 2021 remains included in both the training and validation sets, there will be strong temporal and spatial correlations with those samples, even if footprints are from different receptor points. It would be preferable for the training/validation set to only use data from 2020.
Figures demonstrating performance (e.g., Figure 5) should also be updated to show only 2020, for consistency, and to minimise the influence of data leakage.

We thank the reviewer for this comment. The training, validation, and test datasets are distinct and share no overlapping receptors. Following standard machine learning practices, we randomly sampled training, validation, and test data from a distribution while ensuring

the sets are mutually exclusive, to maintain statistical consistency and prevent data leakage.

Figure 5 shows footprints computed by in-sample FootNet, which is trained on data from both 2020 and 2021. We have now added a supplemental figure (S7) showing footprints computed by out-of-sample FootNet for receptors sampled in 2020, while it was trained on footprints from 2021. We have also updated the Figure 5 caption.

Figure 5: Figure S7 shows a similar comparison for out-of-sample FootNet-predicted footprints for the year 2020.

[Figure]

Figure S7: Footprints computed by out-of-sample FootNet for receptors sampled in year 2020. Out-of-sample FootNet was trained on footprints from 2021.

3. Clarification of the advance claimed by the meteorological dataset comparison (GFS vs HRRR).

The revision has added some minor statements to say that GFS and HRRR differ, and I think that it's useful that the authors have demonstrated that their system is relatively insensitive to the meteorological analysis dataset that was used. However, to my mind, the manuscript still over-sells what this comparison demonstrates. It seems to me that this comparison primarily shows that these analysed meteorological products are very similar to one-another. If they weren't, the emulation wouldn't work, since meteorology is the only input.

To address this comment, all that is needed is needed is for statements such as "despite being trained on HRRR, FootNet accurately predicts footprints using GFS" should be softened or not claimed as a particular advance of Footnet itself (e.g., personally, I wouldn't claim this as a major finding in the abstract).

We thank the reviewer for this comment. As mentioned in our previous response, demonstrating that FootNet can perform well using GFS winds is an important extension to using this model outside CONUS, as HRRR is only available over CONUS. A common question we receive when presenting this work is: *"Can you use other meteorology?"* This comparison clearly demonstrates that we can. As such, it is important to keep this comparison in the text for practitioners who would use FootNet. We have now updated the text in the manuscript.

Line 8: We show that it produces consistent source-receptor relationships when driven by GFS meteorology, even though it was trained with HRRR inputs.

4. Further clarification that Footnet has not "learned the physics"
Related to Point 3, and despite the authors' agreement with similar comments from the first round of reviews, several statements still imply that FootNet has learned "the underlying physics," when in reality it has learned a statistical relationship between meteorology and footprints. Phrases such as "learns the underlying physical relationship" or "learns the fundamental relationship" should be revised to avoid suggesting that physics-informed ML techniques were used. Removing the word "physical", or "fundamental" in these contexts would resolve this concern.

While we agree that stating the model has "learned the physics" is confusing, we have clarified the text to state that the model has learned the underlying physical relationship linking the footprints to the meteorology. Further, as mentioned in our previous response, we enforce a penalty to conserve mass. This is exactly what a physics-informed neural network does. We prefer to leave the language as is because this is supported by the methodology, feature importance, and results.

We have gone back through the manuscript to ensure the language is consistent with the phrasing *"learned the underlying physical relationship"* rather than the former statement: *"learned the physics"*. Again, we feel that this phrasing is supported by the methodology, feature importance, and results.

Minor comments:
Figure 1: The legend now explains the coloring, but the notation for the convolution operations remains unclear. A concise description (e.g., "applied convolution kernel size") would help.

We have now added the convolution kernel size to the Figure 1 caption.

Figure 1: We applied a 3x3 convolution kernel on the convolution layers.

Figure 2: Please clarify what the blue boxes inside Domain B represent.

We thank the reviewer for this suggestion. We have now added the description of the blue boundaries inside Domain B.

Figure 2: The blue boundaries in Domain B show counties in the Barnett Shale basin.

Figure 4 mixes the terms percentage footprint mass difference, percentage footprint sum difference, and normalized footprint mass error. Please use one consistent term and explain precisely what quantity is being plotted.

We thank the reviewer for pointing this out. We now use "percentage footprint sum difference" throughout the manuscript.

Line 94 still contains the term pseudo-observation. This should be replaced with "receptor" or "footprint," consistent with the manuscript revisions.

We thank the reviewer for this comment. We have now updated the manuscript.

Line 92: Each receptor was simulated using STILT(Lin et al., 2003; Fasoli et al., 2018) for surface footprints and X-STILT (Wu et al., 2018) for column-averaged footprints, using NOAA High Resolution Rapid Refresh (HRRR) meteorology at 3 km resolution regridded to 1 km.

Line 129: The text states that the full training dataset is essential. Can you provide justification that a reduced dataset (e.g., 100,000 samples) is insufficient?

We thank the reviewer for this comment. While validation loss begins to plateau beyond approximately 100,000 samples, we found that models trained on smaller subsets exhibited degraded performance in complex terrain and coastal regions. Using 500,000 samples ensures uniform skill across the diverse meteorological and geographic conditions represented within CONUS.

As an example of this, the model trained using only data from 2021 has 170,000 samples. This model exceeds the 100,000 sample criteria we identified and performs well. This is a confirmation of the analysis mentioned. More training data is always preferable, but 100,000 samples *should* be sufficient for training a FootNet model. We have now updated the text.

Line 124: For generalizable inference with uniform skill across the diverse meteorological and geographic conditions represented within CONUS, the full training set remains essential.